# Recent upper Arctic Ocean warming expedited by summertime atmospheric processes

Zhe Li [1], Qinghua Ding [1✉], Michael Steele[2] & Axel Schweiger [2]

The observed upper (0–50 m) Arctic Ocean warming since 1979 has been primarily attributed to anthropogenically driven changes in the high latitudes. Here, using both observational and modeling analyses, we demonstrate that a multiyear trend in the summertime large-scale atmospheric circulation, which we ascribe to internal variability, has played an important role in upper ocean warming in summer and fall over the past four decades due to sea ice-albedo effect induced by atmospheric dynamics. Nudging experiments in which the wind fields are constrained toward the observed state support this mechanism and suggest that the internal variability contribution to recent upper Arctic Ocean warming accounts for up to one quarter of warming over the past four decades and up to 60% of warming from 2000 to 2018. This suggests that climate models need to replicate this important internal process in order to realistically simulate Arctic Ocean temperature variability and trends.

[1] Department of Geography, and Earth Research Institute, University of California, Santa Barbara, Santa Barbara, CA, USA. [2] Polar Science Center, Applied Physics Laboratory, University of Washington, Seattle, WA, USA. ✉email: Qinghua@ucsb.edu

Recent global warming fueled by increasing anthropogenic greenhouse gases is most prominent in the Arctic with significant atmospheric[1–5] and oceanic warming[6–13] and pronounced sea ice and land ice melting[14–18]. Warming of the upper ocean in the Arctic is contributing to sea ice loss[7,11,19–22] and changes of ocean circulation[13,23]. However, our understanding of Arctic upper ocean temperature variability in the past decades and its main drivers remains limited, with previous studies mainly focusing on two processes. The primary one is due to recent sea ice reduction, which allows the ocean to gain more heat[11,12,24,25]. The secondary one involves ocean advection with heat transported into the Arctic Ocean by the time-mean northward-flowing currents[7,26–29].

While the above processes have been extensively examined in the context of anthropogenic warming, the role of internal variability in Arctic Ocean warming is unclear. In the past decades, a strengthened cyclonic oceanic circulation in the Eurasian sector has been observed[30–32] as well as a stronger Beaufort Gyre in the American sector[11,31–33]. However, it is unknown to what degree these are the result of anthropogenic forcing, internal variability, or a combination of both. In particular, the Arctic has exhibited a trend toward higher summertime pressure anomalies since 1979. One explanation links these anomalies to a tele-connection from the tropics which propagates northward via an atmospheric wave train, exerting a warming effect that melts sea ice by regulating temperature, humidity, clouds, and downward longwave radiation in the Arctic atmosphere[34,35]. This process arises from internal variability and explains 40% of the trend in sea ice loss in September since the 1980s[36]. It is reasonable to expect that this process also has an impact on upper ocean temperature either via the ice-albedo effect or via potential impacts on the northward transport of oceanic heat. However, the detailed processes linking this atmospheric internal variability with upper ocean temperature remain unexamined. Because interactions between anthropogenically forced and internally generated processes and feedbacks are complex, it is often difficult to identify a clear cause-and-effect relationship. Given the importance of upper ocean temperature in stabilizing and shaping the high-latitude climate in the Arctic[37,38], a better understanding of the relative roles of each forcing in recent Arctic Ocean warming is desirable.

In this work, we investigate how atmospheric internal variability has regulated upper ocean temperature in the melting season over the last four decades. We seek a physical understanding of the underlying mechanism of this atmosphere-ocean interaction and a quantification of its contribution to the recent warming compared with that due to anthropogenic forcing.

## Results

**Observed linkage between atmospheric circulation and upper ocean warming.** The Arctic Ocean exhibits strong warming trends and year-to-year variability of the upper 50 m in the last four decades in summer and fall[10] (Fig. 1a). This layer, defined as the upper ocean in this study, resides above the Pacific Waters (PW) located between 50 and 150 m depth and Atlantic Waters (AW) located between 200 and 800 m. Thus, its temperature variability impacts the overlying sea ice and the efficiency of the heat exchange between the ocean and atmosphere. To understand upper ocean temperature variability related to summertime atmospheric and sea ice processes, we focus on the area confined by the long-term mean (1979–2018) June–July–August (JJA) Arctic sea ice extent (approximated by black line in Fig. 1b). Within this area, the upper ocean is fully covered by sea ice for large parts of the year but has some exposure to the atmosphere from June to October with the maximum in September when the

sea ice reaches its minimum extent. Thus, an interaction between the atmosphere and the upper ocean is expected during these ice-free months.

Associated with recent atmospheric warming trends that are apparent throughout the year (Fig. 1e), ocean warming trends over the upper 150 m feature a tilted downward intrusion starting from June to August at the surface and propagating downward toward 50 m by fall (i.e, by September–October–November, or SON; Fig. 1f). This downward heat transfer suggests that recent fall upper ocean warming (Fig. 1a) originates from more absorption of heat at the surface in summer (i.e, JJA) because the sea ice-albedo effect which is more efficient in summer allows stronger oceanic uptake of solar radiation during ice melting seasons. This connection operates well on interannual time scales, with the causal direction examined by a lead-lag relationship between JJA Pan-Arctic tropospheric (surface to 300 hPa average) air temperature and ocean temperature in each month and depth (Fig. 1g, h). It is clear that JJA atmospheric warming significantly precedes upper ocean warming from early summer to the following fall and even winter since it takes time to melt sea ice and then warm the ocean due to the larger heat capacities of ocean and sea ice. This calculation suggests that atmospheric forcing drives ocean warming rather than the reverse in summer.

This subsurface fall warming is primarily confined to the Beaufort, Chukchi, and East Siberian Seas (hereafter collectively referred to the "Pacific Peripheral Seas Sector", or PPSS) and the Laptev, Kara, Barents, Norwegian, and Greenland Seas (hereafter referred to as the "Atlantic Peripheral Seas Sector", or APSS; Fig. 1c). The former is the area where the most pronounced sea ice decline in the melting season has been observed since 1979 (Fig. 1c). Concomitant with the trends in upper ocean warming is a trend in the atmospheric upper tropospheric circulation: for example, geopotential height at the tropopause at 300 hPa (Z300: its variability is a measure of temperature variations of the entire air column below 300 hPa; a higher Z300 also means that the circulation changes toward a pattern with stronger anticyclonic movement in the Northern Hemisphere) has been rising over northeastern Canada and Greenland (Fig. 1d). The calculation using the Arctic Ocean domain average variables suggests that the SON upper (0–50 m average) ocean temperature (using reanalysis data; see Methods) has a close association with summertime (i.e, JJA) values of both Z300 and tropospheric air temperature (Table 1). A similar but slightly weaker relationship is observed between the domain average JJA upper ocean temperature with the simultaneous domain average Z300 and tropospheric air temperature (Table 1). SON oceanic warming may reflect an accumulation of changes over JJA due to the larger heat capacities of ocean and sea ice. Thus, the JJA atmosphere—SON ocean connection becomes more significant than the simultaneous connection and the dynamics of this lag relationship will be the focus of this study.

To illustrate the lag process that links the JJA atmospheric circulation pattern to SON upper ocean temperature change, we correlate the domain average SON upper ocean temperature with the spatial fields of JJA Z300 and tropospheric air temperature in the Arctic respectively (Supplementary Fig. 1a, b). An Arctic summer with higher-than-average Z300 and warmer tropospheric air temperature centered over Greenland and the Arctic Ocean appears to consistently precede a warmer-than-average upper ocean in the Arctic in SON (Supplementary Fig. 1a, b). Importantly, the pattern derived from the detrended variables exhibits a very similar structure to the one obtained using raw data, suggesting that the impacts of the JJA atmospheric circulation on the SON upper ocean temperature change also exist on interannual and longer time scales. This domain average SON upper ocean temperature related circulation pattern shows a

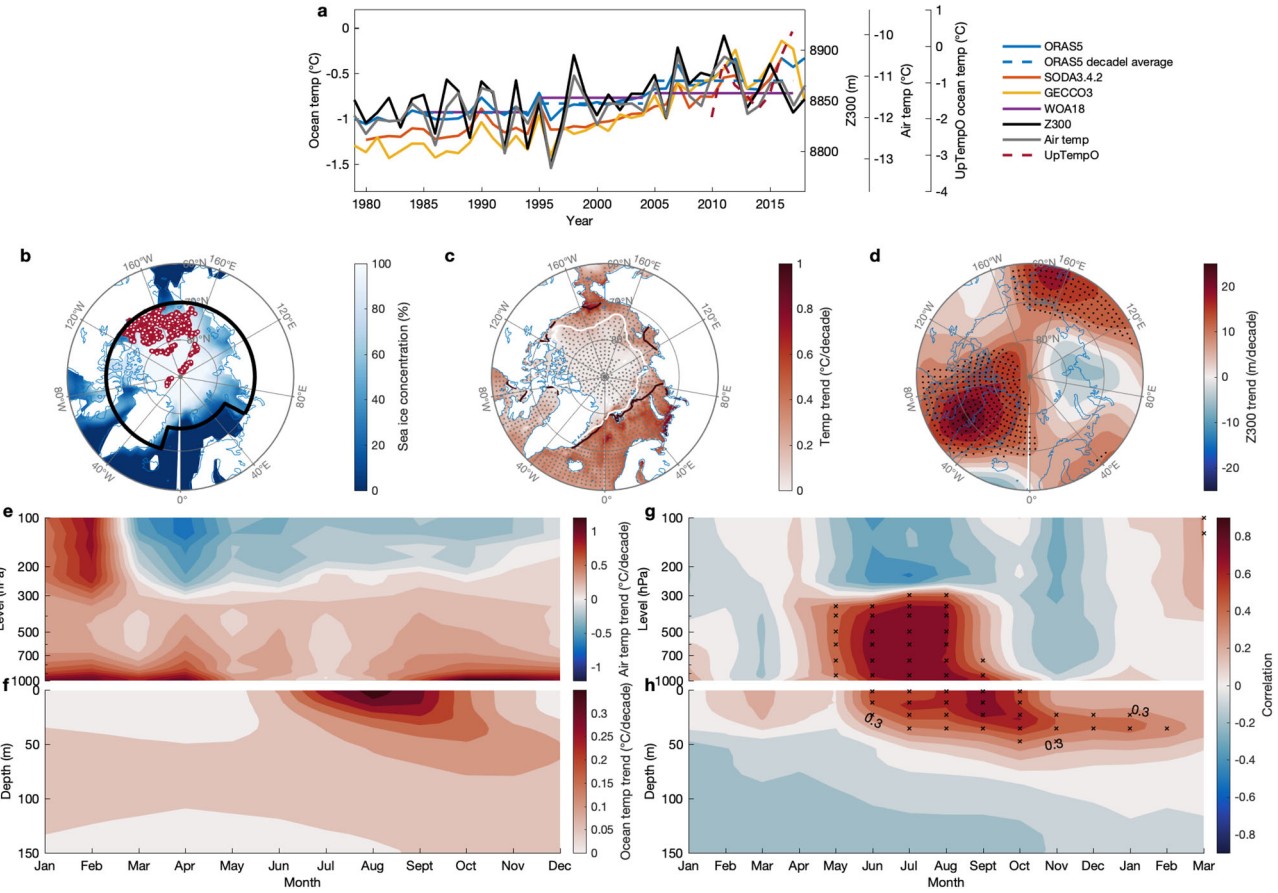

**Fig. 1 Relationship between the summer large-scale atmospheric circulation and upper ocean temperature in fall. a** The Arctic Ocean domain-average time series for SON upper (0–50 m average) ocean temperature (°C) using three different reanalysis data (ORAS5, SODA3.4.2, and GECCO3) and observation data (WOA18), JJA Z300 (m), and JJA tropospheric (surface to 300 hPa average) air temperature (°C) over the region circled by the black contour in **b** using the ERA5 reanalysis, and upper (0–50 m) ocean temperature (°C) using UpTempO buoy data (marked by red dots in **b**). **b** Climatology of JJA sea ice concentration from the National Snow and Ice Data Center (NSIDC) Nimbus-7 SMMR and DMSP SSM/I-SSMIS passive microwave monthly sea-ice product version 1 for the period 1979–2018. The area enclosed by the solid black line in **b** indicates the domain used for the following calculations. Red dots are UpTempO buoy data positions. **c** Linear trend (°C per decade) of SON upper ocean temperature from the ORAS5 reanalysis (1979–2018). September sea ice edges for the periods 1979–1988 mean (the first 10 years) and 2009–2018 mean (the last 10 years) are shown by the black and white contours, respectively. **d** Linear trend (m per decade) of JJA Z300 from the ERA5 reanalysis (1979–2018). Monthly cross-section of the linear trend (°C per decade) of domain-average air temperature over 1000–100 hPa from the ERA5 reanalysis (1979–2018) in **e**, and ocean temperature (°C per decade) over 0–150 m depth from the ORAS5 reanalysis (1979–2018) in **f**. Lead-Lag correlations of JJA domain-average tropospheric air temperature (gray line in **a**) with domain-average air temperature in **g**, and with domain-average ocean temperature in **h**, for each month (the last 3 months: Jan-Feb–Mar are the months in the next year) and layer (atmosphere: from 1000 to100 hPa; ocean: from 0 to 150 m) for the period 1979–2018. All linear trends are removed in calculating the correlations in **g**, **h**. Black and gray stippling in all plots indicates statistically significant correlations or trends at the 95% confidence level.

strong similarity with the linear trend of JJA Z300 field in the past 40 years and therefore upper ocean warming appears to be driven by the atmospheric circulation trend in JJA. Figure 1g, h also indicate that atmospheric warming in JJA influences the ocean surface at zero lag, and influences deeper layers later i.e., in SON. A more rigorous examination of the JJA atmosphere—SON ocean coupling is performed through maximum covariance analysis (MCA) between the JJA atmospheric circulation and SON upper ocean temperature in the Arctic to investigate whether the links between atmosphere and ocean are tied to fundamental modes of variability with a spatially coherent structure (Supplementary Fig. 2). These collectively suggest that a portion of the SON upper ocean warming trend in the past decades results from JJA atmospheric circulation variability on an interdecadal time scale.

**Mechanisms linking JJA atmospheric changes with SON upper ocean temperature.** What processes link JJA atmospheric

## Table 1 Correlations between the summer large-scale atmospheric circulation and upper ocean temperature in fall and summer.

|  | JJA Z300 | JJA tropospheric air temp |
|---|---|---|
| **SON upper ocean temp** |  |  |
| Corr (with trend) | 0.56 | 0.74 |
| Corr (without trend) | 0.48 | 0.66 |
| **JJA upper ocean temp** |  |  |
| Corr (with trend) | 0.53 | 0.69 |
| Corr (without trend) | 0.45 | 0.57 |

Correlations of JJA/SON Arctic Ocean domain-average upper (0–50 m average) ocean temperature with JJA domain-average Z300 and tropospheric (surface to 300 hPa average) air temperature with trends and without trends respectively.

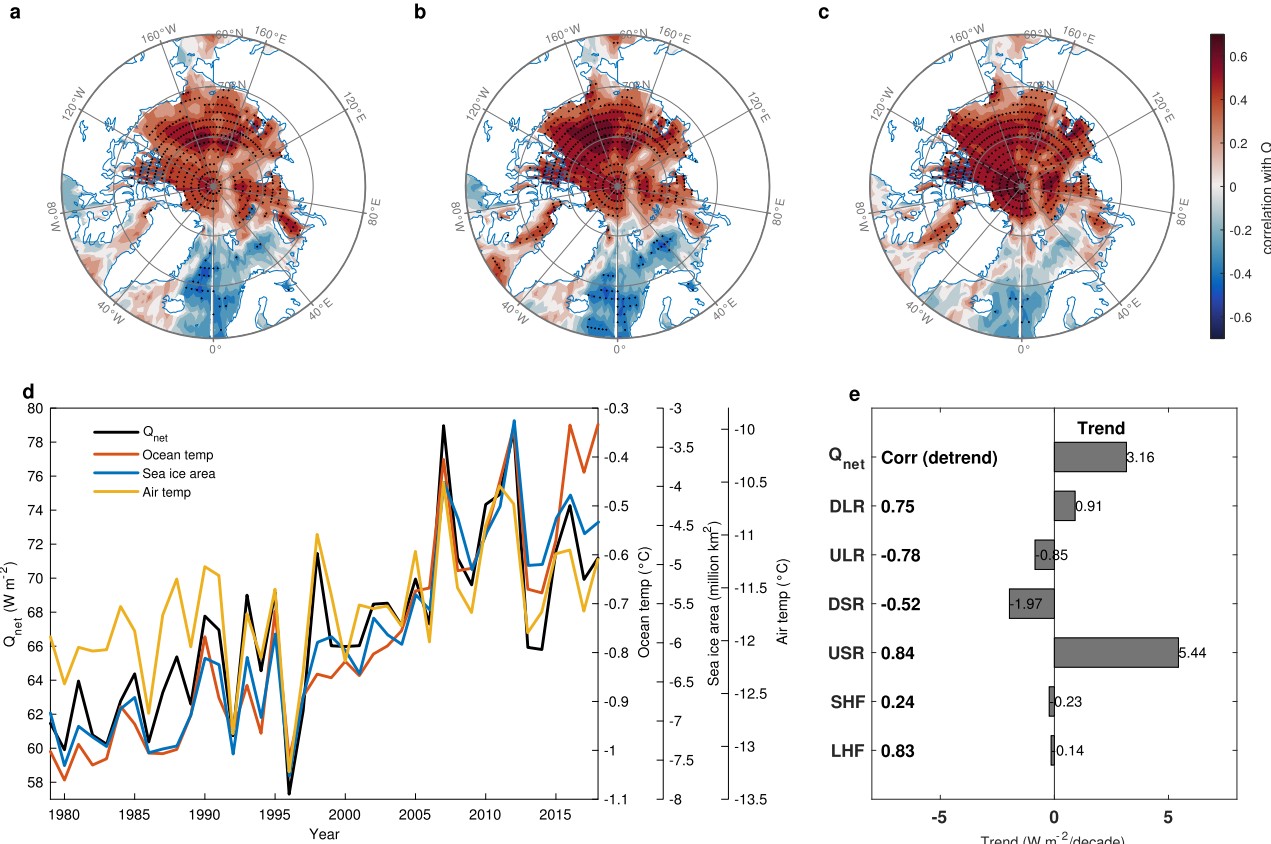

**Fig. 2 Important role of radiation fluxes in linking atmosphere, sea ice, and ocean.** Correlations of 1979–2018 ERA5 reanalysis JJA net downward heat flux ($Q_{net}$) field with three time series, i.e., **a** SON domain-average upper (0–50 m average) ocean temperature, **b** September sea ice area (SIA; average over 70°–90°N, 0–360°, the sign of SIA is reversed for simplicity of comparison with plots **a**, **c**), and **c** JJA domain-average tropospheric (surface to 300 hPa average) air temperature. **d** Domain-average time series for JJA $Q_{net}$ (Wm$^{-2}$), SON upper ocean temperature (°C), September SIA (million km$^2$, the sign of SIA is reversed), and JJA tropospheric air temperature (°C) from 1979 to 2018. **e** Linear trends (Wm$^{-2}$ per decade) of JJA domain-averages of $Q_{net}$, downwelling longwave radiation (DLR), upwelling longwave radiation (ULR), downwelling shortwave radiation (DSR), upwelling shortwave radiation (USR), sensible heat flux (SHF), and latent heat flux (LHF) from the ERA5 reanalysis (1979–2018). All radiative flux variables are positive downward. Correlations of JJA domain-average $Q_{net}$ with JJA domain-averages of DLR, ULR, DSR, USR, SHF, and LHF in the ERA5 reanalysis for the period 1979–2018 are shown in bold next to the y-axis. All linear trends are removed in calculating the correlations in **a**–**c** and **e**. Black stippling in all plots indicates statistically significant correlations at the 95% confidence level.

warming to SON upper ocean warming beneath? Ding et al.[39] showed that a JJA high pressure anomaly can melt summer sea ice through increased downwelling longwave radiation (DLR) at the surface. To investigate how a similar mechanism affects upper ocean temperature, we examine the surface energy transfer between the atmosphere and the ocean. Arctic domain averages of SON upper ocean temperature, September sea ice area (SIA), and JJA tropospheric air temperature are highly correlated with JJA net downward heat flux ($Q_{net}$), all of which exhibit a patch of significant correlations over the Arctic Ocean (Fig. 2a–c, positive for temperature (Fig. 2a, c) and negative for SIA (Fig. 2b)). Domain averages of ocean temperature, SIA, and air temperature co-vary with $Q_{net}$ with detrended correlations ranging from 0.73 to 0.88 (Fig. 2d). This co-variability suggests that atmospheric warming during JJA first melts sea ice, and then warms the resulting open water in the following months.

Next, we examine the trends and correlations between domain average of JJA $Q_{net}$ with the individual component fluxes (see Methods; Eq. 1) over the past 40 years. The increasing trend in $Q_{net}$ is mostly due to reduced upwelling shortwave radiation (USR), and secondarily to increased fluxes in DLR at the surface (Fig. 2e). Further, DLR and USR are two major contributors in determining $Q_{net}$ at the surface on both interannual and interdecadal time scales (Fig. 2d, e). As the surface albedo

decreases with sea ice melt, more solar radiation is absorbed by the darker ocean[35,40,41]. Downwelling shortwave radiation (DSR) decreases substantially over the 40 years, but this is a secondary effect resulting from the reduction of multiple reflection between a shrinking sea ice coverage and clouds[42–44]. Thus, DLR and USR serve as the key fluxes to link JJA air temperature and SON upper ocean temperature in the Arctic.

**Wind-nudging experiments using CESM.** To provide additional evidence that changes in atmospheric circulation are driving the rise of upper ocean temperature through an adiabatic warming process, we conduct a set of nudging experiments to quantify the effect of the atmospheric circulation on upper ocean temperature in the Arctic. In this experiment we nudge the winds of the Community Climate System Model 1[45] (CESM1, which provides a nudging capability) to reanalysis while anthropogenic forcing is fixed at the level of year 2000 ($CO_2$ = 367 ppm), which is very close to the observed mean $CO_2$ concentration over the past 40 years ($CO_2$ = 369 ppm; see Methods). The goal of this experiment is to assess the contribution of wind forcing on sea ice melting and upper ocean warming by comparing the nudging experiment with the historical simulations of the same model and the observational evidence. Since the same model is used in both the

historical and nudging experiments, the comparison of the two sets of experiments (CESM Large Ensemble (CESM-LEN) Project[46] vs. nudging experiments) sheds light on the respective role of winds and anthropogenic forcing in recent changes of upper ocean temperature. First, we examine the response of CESM1 to anthropogenic forcing by examining the 40-member ensemble mean of the historical simulation. The 40-member ensemble is considered sufficient to largely remove the effect of internal variability and thereby only reflects the external forcing. We examine upper tropospheric (300 hPa) winds as an indicator of the larger scale circulation. Unlike the observed upper tropospheric wind trend in ERA5, the wind trend due to anthropogenic forcing is very weak and only accounts for a small part of observed trends (Supplementary Fig. 3). This suggests that the observed upper air wind trend in the past four decades is primarily due to internal variability of the climate system. The nudging experiment consists of five 40-yr historical runs from 1979 to 2018, in which simulated winds within the Arctic (north of 60°N) are nudged to the corresponding 6-hourly ERA5 winds (see Methods). The five members are initiated with different atmosphere, sea ice and oceanic conditions on 1979/1/1 (see Methods) and the ensemble mean of the five realizations is analyzed hereafter to remove impacts of initial conditions in the simulations. The climatology of sea ice concentration, ocean temperature, and salinity in the Arctic in the ensemble mean of these 40-yr nudging runs exhibits roughly similar patterns and magnitude as the ORAS5 reanalysis (Supplementary Figs. 4, 5; See Methods), which gives us confidence that the model has sufficient skill to simulate the mean state in the Arctic Ocean.

The simulated spatial pattern of upper ocean temperature trend is similar to that in ORAS5 in the PPSS from 1979 to 2018 and the pattern in the PPSS and APSS for the 2000–2018 period although the temperature increases are slightly weaker (Fig. 3c–f). It is particularly noted that the warming in the Barents Sea in the nudging simulations bears strong resemblance to ORAS5 for the 2000–2018 period (Fig. 3e, f), with the spatial correlation coefficient between these two trend patterns (Fig. 3e, f) within the Arctic (north of 70°N) reaching 0.77. The simulated domain average SON upper ocean temperature shows a highly correlated temporal variation with ORAS5 (for the period 1979–2018: $r = 0.63$ with trend, $r = 0.67$ without trend; for the period 2000–2018: $r = 0.91$ with trend, $r = 0.80$ without trend) on both interannual and interdecadal time scales (Fig. 3a). This suggests that the wind-driven circulation change indeed plays an important role in upper ocean warming in the Arctic. The mean trend in the simulated upper ocean temperature in the five nudging runs is 0.04 °C per decade, while that in the ensemble mean of the 40 CESM-LEN members is 0.09 °C per decade. The combined upper ocean temperature trend due to the two forcings is 0.13 °C per decade, which is slightly lower than the trend of 0.17 °C per decade in ORAS5 over the last 40 years, suggesting that the sum of these two forcings can explain most of SON upper ocean warming. Based on their contributions to the warming in ORAS5 (0.04/0.17 and 0.09/0.17), we estimate that the internal, wind-driven variability accounts for 24% of upper ocean warming while anthropogenic forcing accounts for 53% of upper ocean warming over the past 40 years. While the wind-driven ocean warming is largely confined to the Chukchi, East Siberian, and Laptev Seas over the 40-year period, substantial warming in the Atlantic sector shows little connection with wind-driven processes. This changes when we perform a similar calculation for the period 2000–2018, when reanalysis wind and ocean data in the Arctic are more reliable[47,48], and upper ocean temperatures warmed significantly. In this case, we find that internal wind-driven variability has become even more important and over the Pan-Arctic Ocean, explaining about 60% of the Pan-Arctic Ocean warming trend from 2000 to 2018. A caveat is that anthropogenic warming likely has an imprint on the reanalysis winds which are used to drive the nudging run, although this part appears to be small (Supplementary Fig. 3). This means that our estimates of the role of internal variability on SON upper ocean warming (24% over 1979–2018 and 60% over 2000–2018) are likely upper bounds.

In order to check that the model is accurately simulating ocean conditions, we next compare the simulated upper ocean temperature trend in the nudging experiments (see Fig. 3g) with that in ORAS5. Comparison with Fig. 1f indicates that the model is capturing the main warming, although its magnitude is only 25% of values in ORAS5. The lagged correlation between simulated domain average JJA tropospheric air temperature and domain average ocean temperature also yields a similar pattern as the counterpart in ORAS5 in Fig. 1h, i.e., significant JJA atmospheric warming precedes the ocean temperature rise that shows a strong downward intrusion into lower layers from June through the following winter (Fig. 3h).

Consistent with observations in ERA5, DLR and USR in both the nudging and historical runs are the two main factors contributing to the positive $Q_{net}$ trend over the past 40 years (Supplementary Fig. 6a). In the nudging runs, wind-driven impacts on DLR and USR are comparable with that due to $CO_2$ forcing, suggesting that large-scale circulation variability plays a similar role as $CO_2$ forcing in triggering the sea ice-albedo effect seen as an increase of USR in Supplementary Fig. 6a. Nudging CESM1 to observed winds yields simulated DLR and USR averages capturing over 50% of the observed variability in those energy balance components, with $r = 0.72$ and 0.74 for DLR and USR respectively (Supplementary Fig. 6b).

To better understand mechanisms contributing to upper oceanic warming and its deepening in our nudging runs, we examine the trends of SON mixed layer depth (MLD) in the Arctic Ocean. The deepening of MLD is observed in both ORAS5 and the nudging runs since 1979, especially for the period from 2000 to 2018 (Supplementary Fig. 7), indicating that the wind-driven vertical mixing can partially explain enhanced downward heat transport in the upper ocean through increasing the MLD. Although the vertical mixing is overestimated in the simulations compared with that in ORAS5 (Supplementary Fig. 7a, d; Fig. 3h), it remains unclear which one is closer to observations since most climate models and reanalyses contain large uncertainties to capture the MLD and its variability in the Arctic[49]. Furthermore, to trace how changes in $Q_{net}$ propagate into the upper ocean where they can contribute to warming, we examine the relationship of $Q_{net}$ with the solar shortwave radiation heat flux penetrating into the ocean boundary layer ($Q_{short\_bl}$; see Methods) in our nudging runs. This layer on average is ~ the upper 30 m of the ocean in the nudging runs. An increase in SON $Q_{short\_bl}$ from 1979 to 2018 is primarily confined to the PPSS (Supplementary Fig. 8a)[50]. High correlations between JJA domain average $Q_{net}$ and $Q_{short\_bl}$ field in JJA exist over the peripheral seas (the Beaufort, Chukchi, East Siberian, Laptev, and Kara Seas) but shift to the entire Arctic Ocean for SON (Supplementary Fig. 8b, c). Figure 3b more clearly reveals the temporal relationship of $Q_{net}$ and $Q_{short\_bl}$, showing that $Q_{net}$ in JJA is highly correlated with simultaneous and subsequent $Q_{short\_bl}$ until November, with the highest correlations when $Q_{net}$ leads $Q_{short\_bl}$ by 1 month. The spatial and temporal lead-lag relationship of $Q_{net}$ and $Q_{short\_bl}$ is what is expected if changes in $Q_{net}$ first drive sea ice loss and subsequent upper ocean warming. This provides the physical mechanism that ties large-scale wind variability and its effect on $Q_{net}$ to upper ocean warming via the ice-albedo effect and the deepening of the MLD.

**Poleward ocean heat transport contribution**. Although the nudging runs well-replicate the upper ocean temperature rise in

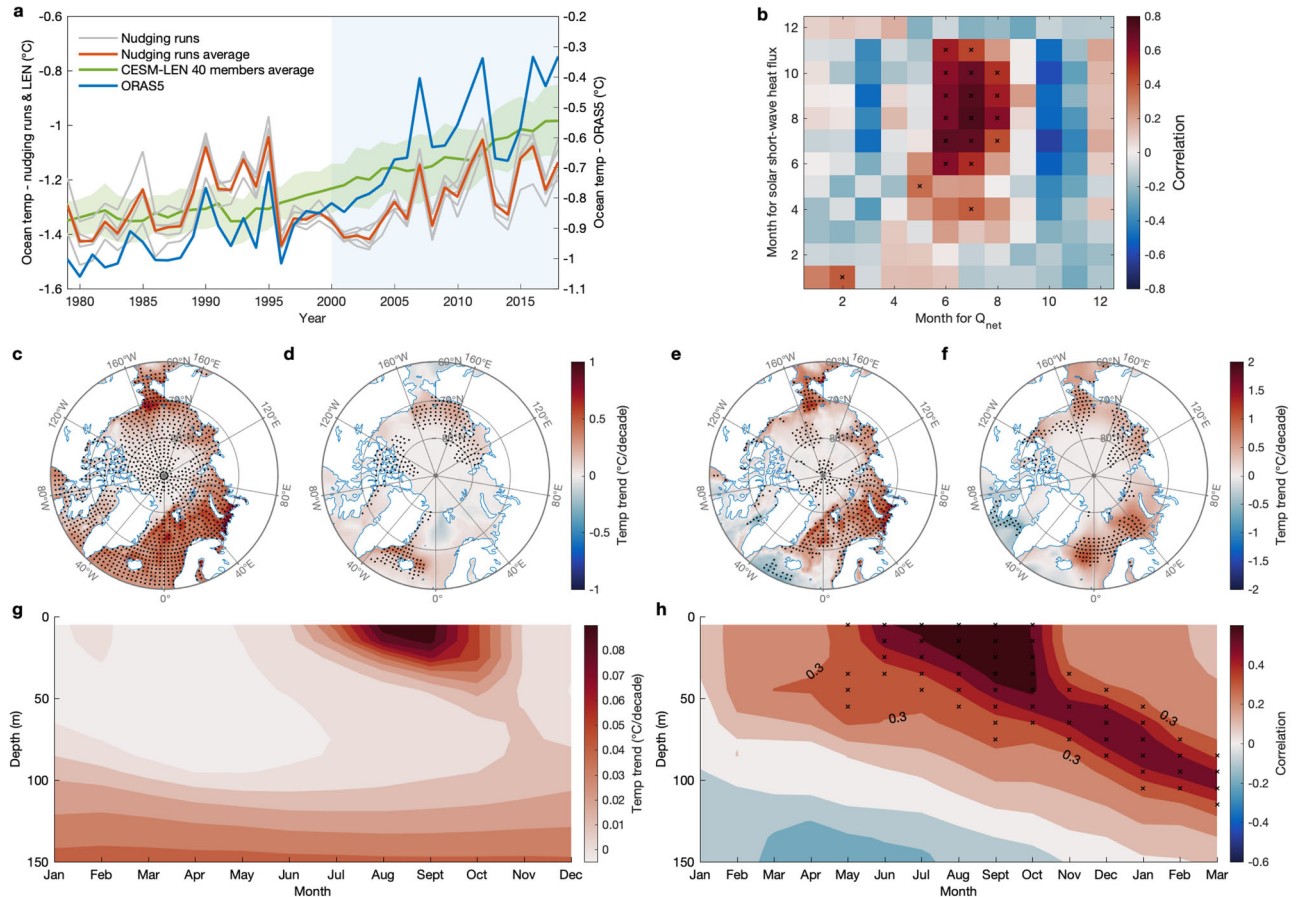

**Fig. 3 Simulated impact of summertime atmospheric processes on Arctic Ocean temperature in the wind-nudging experiments. a** SON Arctic Ocean domain-average upper (0–50 m average) ocean temperature (°C) in the five wind-nudging experiments (gray line) from 1979 to 2018, CESM–LEN 40 members average (green line), and ORAS5 reanalysis (blue line) from 1979 to 2018. The red line represents the ensemble average of the five wind-nudging experiments. The green shading represents the one standard deviation spread of the all CESM-LEN 40 members away from the ensemble mean. **b** Correlations of monthly domain-average solar shortwave heat flux absorbed in the ocean boundary layer with monthly domain-average net heat flux ($Q_{net}$) at the surface for any pair of 2 months for the period 1979–2018 from the ensemble average of the five wind-nudging experiments. Linear trend (°C per decade) of upper ocean temperature from the ORAS5 reanalysis in **c**, and from the ensemble average of the five wind-nudging experiments in **d** for the period 1979–2018. **e**, **f** The same as **c**, **d**, but for the period 2000–2018. **g** Monthly cross-section of the linear trend (°C per decade) of domain-average ocean temperature over 0–150 m depth for the period 1979–2018 from the ensemble average of the five wind-nudging experiments. **h** Correlation of JJA domain-average tropospheric air temperature with domain-average ocean temperature, for each month (the last 3 months: Jan–Feb–Mar are the months in the next year) and depth (from 0 to 150 m) for the period 1979–2018 from the ensemble average of the five wind-nudging experiments. All linear trends are removed in calculating the correlations in **b**, **h**. Black stippling in all plots indicates statistically significant correlations or trends at the 95% confidence level.

the Arctic, especially over the PPSS, model vs. reanalysis (ORAS5) differences remain (e.g., Fig. 3c, d). This indicates that some additional factors not directly captured by atmospheric wind forcing may play a role in driving changes in the Arctic, especially over the APSS. Poleward ocean heat transport (POHT) is in part driven by winds but are also affected by large-scale ocean dynamics that are not directly tied to winds (or at least not at the time scales considered here). We consider POHT in the upper 50 m through two separate gates into the Arctic Ocean, a Pacific Gate measuring heat inflow through the Bering Strait and an Atlantic Gate measuring net heat inflow from the Nordic Seas (Fig. 4a). In this section, we explore the role of POHT on upper ocean warming, although we do not provide a quantitative "variance explained" analysis as in previous sections. This is because an exercise that would involve ocean state nudging is beyond the scope of this study.

SON upper 50 m POHT through the Atlantic Gate derived from ORAS5 shows an upward trend since 2000 (Fig. 4a), especially via the branch of that through the Barents Sea

(Supplementary Fig. 9b), and is strongly correlated with SON upper ocean temperature on both interannual and interdecadal time scales for the period 1979–2018 ($r = 0.80$ with trend; $r = 0.58$ without trend). Correlating the POHT time series with upper ocean temperature field shows that the variability of POHT through the Atlantic Gate in SON strongly affects the Barents and Kara Seas (Fig. 4b). The cause of the high correlations in the parts of the central Arctic Ocean is puzzling. One possible reason is that an air-sea heat flux exchange is able to quickly take the warm signal from the Barents and Kara Seas to the central Arctic Ocean via the atmosphere[51]. This simultaneous connection may also result from some influences in preceding seasons that can regulate upper ocean temperature in both the central Arctic Ocean and the Barents and Kara Seas. In contrast, POHT through the Pacific Gate has an increasing trend as well (Fig. 4a) but only affects the Chukchi Sea (Fig. 4c).

The variability of simulated SON upper 50 m POHT through the Pacific Gate in the wind-nudging runs very successfully capture the counterpart in ORAS5 (Fig. 4e; $r = 0.9$ with trends,

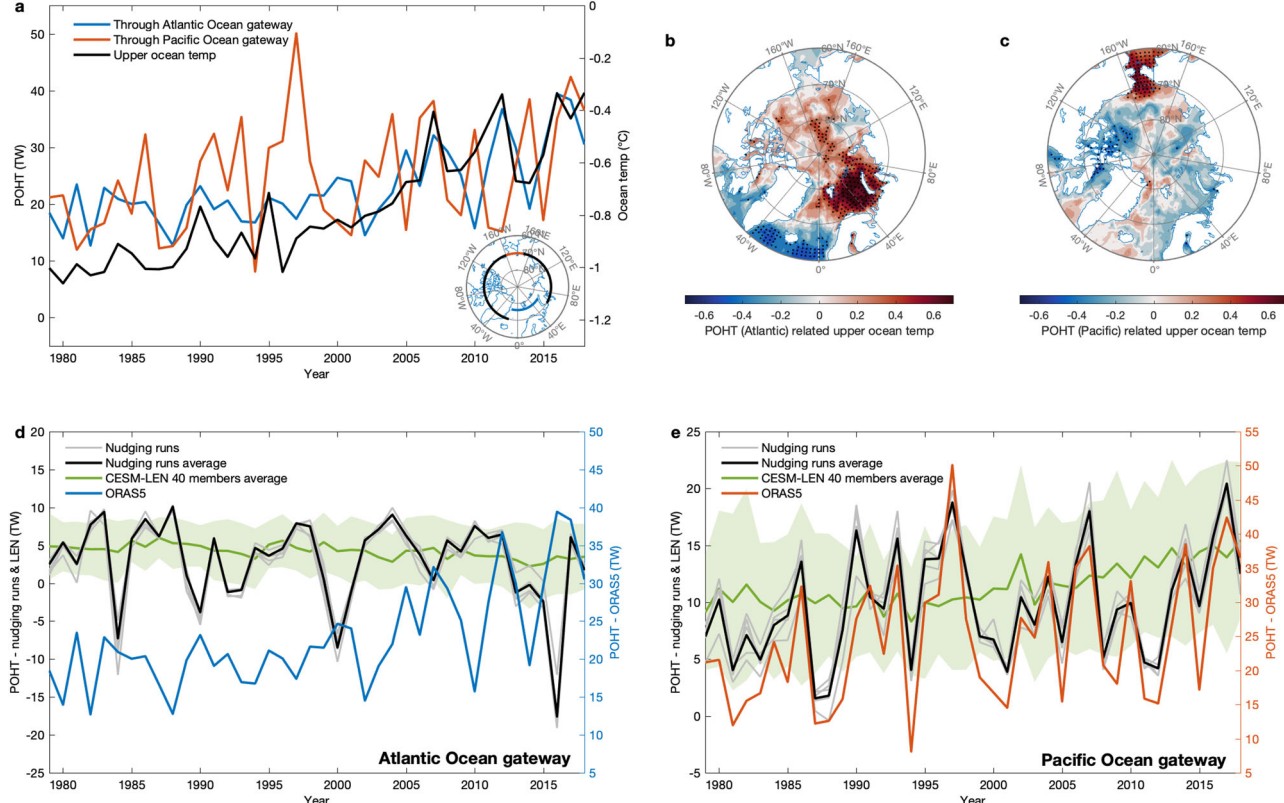

**Fig. 4 Poleward ocean heat transport through the Atlantic and Pacific Ocean gateways in fall. a** SON poleward ocean heat transport (POHT; unit: TW, $1\,TW = 10^{12}$ W) within upper 50 m through the Atlantic Ocean gateway (blue line) and Pacific Ocean gateway (red line), and SON domain-average upper (0–50 m average) ocean temperature (°C, black line) using the ORAS5 reanalysis for the period 1979–2018. The Atlantic and Pacific Ocean gateways are indicated by the blue and red contours in the stereographic projection (lower right in **a**). Correlations of 1979–2018 SON upper ocean temperature field with two time series, i.e., **b** SON upper 50 m POHT through the Atlantic Ocean gateway in **a** (blue line), **c** SON upper 50 m POHT through the Pacific Ocean gateway in **a** (red line). SON upper 50 m POHT (TW) from 1979 to 2018 through the Atlantic Ocean gateway in **d**, and through the Pacific Ocean gateway in **e** from the five wind-nudging experiments (gray lines, and ensemble average in black line), CESM-LEN 40 members average (green line), and the ORAS5 reanalysis (blue line for the Atlantic Ocean gateway, red line for the Pacific Ocean gateway). The green shading represents the one standard deviation spread of the all CESM-LEN 40 members away from the ensemble mean. All linear trends are removed in calculating the correlations in **b**, **c**. Black stippling in all plots indicates statistically significant correlations at the 95% confidence level. POHT through the Atlantic Ocean and Pacific Ocean gateways are calculated using different reference temperatures (0 °C for the Pacific Ocean gateway and −1.9 °C for the Atlantic Ocean gateway, respectively).

$r = 0.89$ without trends), but this is not the case for the Atlantic Gate (Fig. 4d and Supplementary Fig. 9). This suggests that winds play an important role in driving POHT through the Bering Strait[52,53]. Nevertheless, wind-driven POHT through the Bering Strait only has a weak correlation with upper ocean temperature in the interior of the basin and appears to have little impact on Pan-Arctic Ocean warming. We also compare SON upper 50 m POHT via the Atlantic Gate in the nudging runs with that in the 40-member ensemble means of the historical simulation in CESM (Fig. 4d, e). None of these capture the increasing trend of POHT through the Atlantic Gate as seen in ORAS5, suggesting that SON upper 50 m POHT changes through the Atlantic Gate are likely determined by more complex factors that are not directly driven by winds and anthropogenic forcing in our model, such as the initial ocean condition, deeper layer oceanic variability, internal oceanic thermohaline variability in the Arctic and heat transport from the lower latitudes where observed winds are not specified[19,54–56]. The role of anthropogenic forcing in contributing to increasing POHT via the Atlantic sector remains an open question since this attribution appears to be sensitive to approaches used to detect this feature[19,57]. Importantly, the fact that our nudging simulations did not capture an upward trend in POHT through the Atlantic Gate further supports our main

finding: namely, the role of our identified mechanism in expediting Arctic Ocean warming via summertime atmospheric processes.

## Discussion

Our study suggests that a portion of upper Arctic Ocean warming over the past few decades can be explained by low-frequency atmospheric variability characterized by a trend toward anomalous anticyclonic circulation over the Arctic Ocean and Greenland. This process produces subsidence and adiabatic warming which acts to warm the atmosphere, melt sea ice, and deepen the ocean mixed layer. The resulting open water warms via shortwave radiation absorption and enhanced vertical mixing. Our nudging experiments confirm that adiabatic dynamical forcing associated with winds in the Arctic is able to explain up to 24% of SON upper ocean warming from 1979 to 2018, which is mostly confined to the Chukchi, East Siberian, and Laptev Seas, and up to ~ 60% of the Pan-Arctic upper ocean warming for the period 2000–2018.

We have previously suggested that internal atmospheric forcing is in part forced by sea surface temperature variability in the tropics via a Rossby wave train[34]. The capability to replicate both local air-ice-ocean coupling as well as an accurate tropical—

Arctic Ocean connection is thus a key model skill. However, simple evaluations conducted in previous studies indicate that some models have trouble replicating both the full strength of the local coupling on interannual time scales and also tropical—Arctic Ocean connections[58,59]. An efficient and well-developed metric, emphasizing a lead-lag connection between JJA atmospheric temperature with SON ocean temperature as we show in Fig. 1f, g, is thus needed to better evaluate the performance of models in representing this Arctic atmosphere-sea ice-ocean coupling and its possible linkage with remote forcing.

In this study, we primarily focus on the surface layer of the Arctic Ocean and local ocean—air coupling through thermodynamical processes. However, the surface heat balance is also influenced by ocean mixing, the deep ocean circulation and heat transport from sub-Arctic Oceans[7,26,27,29]. In particular, the barotropic feature of the POHT through the Atlantic Gate (see Methods) requires an integrated view of heat transport throughout the whole depth. Other factors, such as water mass exchanges between the Arctic and Atlantic/Pacific Oceans[25], and freshwater storage changes[60] become more important when we shift our focus to ocean lateral heat flux convergence and the deeper layers of the Arctic Ocean. Moreover, the ocean mixing is not only sensitive to surface winds[61] and Ekman convergence and pumping in the surface layer around the central Beaufort Gyre[62], but also regulated by brine rejection[63]. Recently, the "Atlantification" process has been suggested to be associated with weakened stratification and increased vertical mixing, which enhances upward heat fluxes to the surface[27,64]. This process may influence upper ocean warming and sea ice melt and may not be well-replicated in our simulations. Further studies are needed to better understand the relationship and interaction between the Arctic Ocean and the internal large-scale atmospheric process described here.

## Methods

### Reanalysis and observation data
In situ observations of the upper ocean temperature are very limited and sparse in the Arctic Ocean compared to the North Atlantic[65], which inhibits our ability to study the multi-decadal large-scale variability of climate diagnostics, such as heat and salinity budgets. However, observational constraints on future anthropogenic warming critically depend on accurate estimates of past ocean temperature change. Reanalyses are another tool that provide multi-variate dynamical consistency in both spatial and temporal dimensions[66]. We primarily use the Ocean Reanalysis System 5 (ORAS5)[67] in this study to investigate the changes in upper ocean temperature and heat transport and their relationship with atmospheric variability over the past four decades. The ORAS5 dataset (the horizontal resolution is about 0.25° × 0.25°) is constructed by the ECMWF global operational ensemble reanalysis system containing an eddy-permitting ocean and sea-ice system and the OCEAN5 system. Reprocessed sea surface temperature from HadISST2 and OSTIA operational, sea-ice concentration from OSTIA and OSTIA operation, in situ temperature/salinity profiles from EN4 with XBT/MBT correction and GTS operational, and satellite sea level anomaly from AVISO DT2014 with revised MDT are assimilated in this system via NEMOVAR[68] using a 5 day assimilation window with a model time step of 1200 s[67]. Global mean sea-level changes are constrained using AVISO DT2015 L4 MSLA and NRT. The atmospheric forcing of ORAS5 is derived from ERA-40 (before 1979), ERA-Interim (1979–2015), and ECMWF NWP (2015–present)[67]. ORAS5 includes five ensemble members and covers the period from 1979 onwards. In this study, the ensemble mean of the five members is analyzed. Although various in situ observations are assimilated in ORAS5, there are only a limited number of temperature/salinity profiles in the EN4 product in the central Arctic Ocean, which may potentially degrade the accuracy of ORAS5 in reflecting real observations over the region.

To evaluate the ORAS5, we compare its upper (0–50 m average) ocean temperature changes in the Arctic Ocean over the period 1979–2018 with other widely used reanalysis products (SODA3.4.2[69] and GECCO3[70]) and observation (WOA18[71]). All ocean temperatures used in the calculations are potential temperatures, and we convert in situ temperature from WOA18 to potential temperature to be consistent with other reanalyses. The trends of upper ocean temperature time series in ORAS5 and SODA3.4.2 are similar with 0.165 °C per decade for ORAS5 and 0.187 °C per decade for SODA3.4.2 (note the shorter time period from 1980 to 2016), both of which are smaller than that of GECCO3 (0.279 °C per decade). However, more important for this study, the variabilities of temperature time series and spatial patterns of upper ocean temperature trends in

the three reanalyses are consistent (Fig. 1a and Supplementary Fig. 10). Comparisons with 'observation' only dataset is complicated by the fact that WOA18 only provides decadal monthly means of temperature for three decades (1985–1994, 1995–2004, 2005–2017), but the decadal average of upper ocean temperature time series derived from ORAS5 over two decadal periods (1985–1994 and 1995–2004) are close to that in WOA18, while it is slightly larger than WOA18 over the period 2005–2017 (Fig. 1a). Spatial patterns of long-term trends of upper ocean temperature derived from reanalyses are similar to WOA18 with pattern correlation coefficients of 0.74 for ORAS5, 0.65 for SODA3.4.2 and 0.64 for GECCO3 (and ORAS5 has the highest correlation), although GECCO3 exhibits stronger warming than WOA18 over the whole basin (Supplementary Fig. 11). In addition, observational data from the UpTempO Buoy Project[72] was used to compare with reanalyses and we find that they well replicate UpTempO upper ocean temperature variability from 2000 to 2017 (Fig. 1a). Based on these evaluations, we believe that ORAS5 is a reliable data source for this study.

We also use monthly circulation, air temperature, and radiation fields from 1979 to 2018 from ERA5 reanalysis data[73]. Monthly sea ice concentration is obtained from Nimbus-7 SSMR and DMSP SSM/I-SSMIS passive microwave data version-1 provided by the National Snow and Ice Data Center (NSIDC)[74].

### Maximum covariance analysis (MCA)
MCA[75,76] is used to determine the dominant covarying patterns of high-latitude atmospheric circulation and upper ocean warming in the Arctic. MCA analysis is applied by using singular value decomposition on the temporal covariance matrix between JJA NH high-latitude (60–90°N) Z300 and SON upper (0–50 m average) ocean temperature over the Arctic (70–90°N). Simply put, MCA analysis can isolate the most coherent pairs of the spatial patterns and identify a linear relationship between two different fields that are most closely coupled. The leading modes show the spatial patterns of the two fields that are optimally coupled. The corresponding squared singular value represents the squared covariance fraction, which indicates the relative importance of that pair of vectors in relationship to the total covariance in the two fields.

### Radiation fluxes
$Q_{net}$ is calculated as the following, with units of $Wm^{-2}$

$$Q_{net} = DLR - ULR + DSR - USR + SHF + LHF \quad (1)$$

where DLR and ULR are downwelling and upwelling longwave radiations, respectively; DSR and USR are downwelling and upwelling shortwave radiations, respectively; SHF is sensible heat flux; LHF is latent heat flux. All radiative flux variables are positive downward and represent heat transferred from the atmosphere to the surface.

### Wind-nudging experiments using CESM1
Because we are interested in the contribution of wind forcing to upper ocean warming in the Arctic in our model simulations, we use 6-hourly ERA5 wind fields for nudging. The experiments consist of five 40-year historical runs from 1979 to 2018 using the CESM1 fully-coupled model[45]. In these runs, Arctic (north of 60°N) atmospheric winds from the surface to the top of the atmosphere in the model are fully nudged to the corresponding 6-hourly ERA5 winds (the relaxation timescale of the nudging is about 6 h) with various different sea ice initial conditions derived from a long (150 year) spin-up simulation, and there is no nudging effect everywhere else. Anthropogenic forcing is held constant in the nudging experiments and spin-up run at the level of year 2000 (i.e. $CO_2$ is set at 367 ppm) so that greenhouse gas concentrations throughout the integration is very close to the observed 40-yr averaged values (i.e. $CO_2$ is set at 369 ppm). To address the issue of "assimilation shock", the tendency of models to equilibrate to imposed winds, we first run a 150-year perpetual simulation with the model continuously nudged to winds in year 1979 in the Arctic (from the surface to the top of atmosphere) and forced by constant anthropogenic forcing from year 2000. In this spin-up, the model takes almost 100 years for the Arctic mean upper ocean temperature to adapt to wind fields and then varies stably around a constant level afterwards (Supplementary Fig. 12). The model states on Jan. 1 of the last 5 years of this spin up are then separately used as initial conditions to reinitiate a set of new five members of 40-yr nudging simulations in which imposed winds in the Arctic are allowed to vary from 1979 to 2018. In this study, we focus on the ensemble mean of these five members to understand wind forcing on ocean temperature and other fields in the Arctic. In addition, one advantage of this set of simulations is that the POP2 of the CESM1 can simulate $Q_{short\_bl}$, which is calculated by the KPP vertical mixing scheme and not available in ORAS5. This variable will tell us how much of incoming shortwave flux at the surface can be absorbed by the whole depth of the ocean surface boundary layer in each oceanic grid based on the fraction of solar shortwave flux penetrating to the bottom of this layer[77].

### The mean sea ice and oceanic states simulated by wind-nudging experiments
Although we are interested in the anomalous response to imposed winds, an examination of the simulated sea ice and oceanic mean states is necessary to ensure that the model can capture similar climatology in ORAS5 to prepare the model to reasonably respond to imposed forcing. We examine the long-term (1979–2018) mean March and September sea ice extent in observations (Supplementary Fig. 4a, b)

and the ensemble mean of five 40-yr nudging experiments (Supplementary Fig. 4c, d). Nudging experiments appear to have a good skill to capture the observed sea ice states over the Arctic, with slightly smaller extent in September than observed. Supplementary Fig. 5 shows a comparison of the long-term mean (1979–2018) February and August ocean temperature and salinity in ORAS5 reanalysis (a, c, e, and g) and wind-nudging experiments (b, d, f, and h) over a vertical cross-section from Alaska to Svalbard via the North Pole. The model captures the inflow of warm, salty AW near Svalbard, although its salinity is too high at the inflow and its depth interval within the Arctic Ocean is deeper than that in ORAS5. A realistic signature of the Beaufort Gyre between Alaska and the North Pole is seen as concave up (i.e., downwelling) isohaline and isothermal contours. Ocean surface summer warming and freshening is also captured in the model. However, deficiencies still remain: One is the absence of a warm summer PW layer in the Canada Basin in the model and the other is the cooler mean state of upper ocean temperature in the model. These are common issues even in some regional models[62].

**Poleward ocean heat transport.** For a given depth layer, POHT is calculated as the cross-section integral of temperature multiplied by meridional ocean velocity, seawater density and the specific heat capacity, with units of Watts:

$$\text{POHT}(V_R, t) = \int_D^0 \int_{west}^{east} V(\theta - \theta_r) dz \cdot dx \cdot C_P \cdot \rho \tag{2}$$

where $D$ is the depth of ocean which we take to be 50 m, $V$ is temperature minus a reference temperature, $C_p = 4200\,\text{J}\,\text{kg}^{-1}\,\text{C}^{-1}$ is the seawater specific heat capacity, and $\rho$ is seawater density. We assume the density of ocean water as a constant $1027\,\text{kg}\,\text{m}^{-3}$ for the Atlantic Gate and $1025\,\text{kg}\,\text{m}^{-3}$ for the Pacific Gate because density has not changed much in reality. As prior works, we use $\theta_r = 0\,°\text{C}$ for the Atlantic Gate[19] and $\theta_r = -1.9\,°\text{C}$ for the Pacific Gate[78–81]. Cross sections are indicated by blue and green contours in the stereographic projection in Fig. 4a for the Atlantic Gate (15°W–60°E longitude at 76°N latitude) and the Pacific Gate (170°E–160°W longitude at 70°N latitude). We also split the POHT through the Atlantic Gate into two branches through the Fram Strait (15°W–15°E longitude at 76°N latitude) and Barents Sea (20°E–60°E longitude at 76°N latitude). The heat inflow in the upper 50 m through the Bering Strait is sufficient to modify the ocean temperature in the upper 50 m because the Bering Strait is only 50 m deep. Over the Atlantic sector, the variability of POHT through the Atlantic Gate from 0–300 m or 0–2500 m is similar to that in the upper 50 m because the AW inflow is largely barotropic (Supplementary Fig. 13). Considering that the upper 50 m ocean temperature is the most sensitive to the heat inflow in the upper 50 m, we primarily focus on the POHT at this level in this study. We also note that there is a mismatch between the magnitudes of simulated POHT and that in ORAS5 through the two gateways (Fig. 4d, e). The main source of this mismatch appears to be the underestimated intensity of poleward volume transport in the model (Supplementary Fig. 14d, e).

**Significance of correlation.** The statistical significance of the correlation coefficient accounts for the autocorrelation in the time series by using an effective sample size $N^*$[82]:

$$N^* = N \frac{1 - r_1 r_2}{1 + r_1 r_2} \tag{3}$$

where $N$ is the number of available time steps and $r_1$ and $r_2$ are lag-one auto-correlation coefficients of each variable. A confidence level of 95% is used in this study to determine the significance of correlations and composites.

## Data availability

All reanalysis data used in this study were obtained from publicly available sources. Simulated global circulation, temperature, sea ice, ocean temperature and currents under anthropogenic forcing were obtained from the LENS archives accessed through the Earth System Grid Federation data portal (http://esgf.llnl.gov). Sea Ice Concentration from Nimbus-7 SMMR and DMSP SSM/I-SSMIS Passive Microwave Data, Version 1 is accessed from NSA DAAC at the National Snow and Ice Data Center at https://nsidc.org/data/nsidc-0051. The ECMWF ERA5 reanalysis product is available at https://www.ecmwf.int/en/forecasts/datasets/reanalysis-datasets/era5. The ECMWF Ocean ReAnalysis System 5 (ORAS5) product is available at https://www.ecmwf.int/en/research/climate-reanalysis/ocean-reanalysis. The SODA3 product is available at https://www.soda.umd.edu/. The GECCO3 ocean synthesis is available through Integrated Climate Data Center (ICDC) at https://icdc.cen.uni-hamburg.de/en/gecco3.html. The UpTempO buoy data is available at http://psc.apl.washington.edu/UpTempO/Data.php and https://arcticdata.io/catalog/view/doi%3A10.18739%2FA2D21RK2N. The Word Ocean Atlas 2018 (WOA18) is available at https://www.ncei.noaa.gov/products/world-ocean-atlas. All data needed to evaluate the conclusions in the paper are present in the paper and/or the Supplementary Materials. The data generated in this study have been deposited in the OSF and can be accessed via https://osf.io/y8j27/. In addition, the CESM wind-nudging experiment raw output is available from the corresponding author upon request.

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

## Acknowledgements

This study was jointly supported by Modeling, Analysis, Predictions and Projections (NA19OAR4310281) and Climate Variability & Predictability (NA18OAR4310424) programs as part of NOAA's Climate Program Office, and NSF's Polar Programs (OPP-1744598). M.S. was supported by NSF grants PLR 1603266 and OPP-1751363 and ONR grant N00014-17-1-2545. A.S. was supported by NSF grants OPP-1744587, PLR-1643436 and NASA grant 80NSSC20K1253. We acknowledge the CESM Large Ensemble Community Project and supercomputing resources provided by NSF/CISL/Yellowstone (https://doi.org/10.5065/D6RX99HX) and CESM Polar Climate Working Group.

## Author contributions

Q.D. conceived the study and conducted the numerical experiments. Z.L. generated the first draft further improved by Q.D., M.S., and A.S. Z.L. led the analyses and evaluation of

results, with assistance from Q.D, M.S., and A.S. M.S. provided particular guidance on ocean observations and ocean model analysis. A.S. provided insight on climate model analysis. All authors equally contributed to the writing and the revision of this article.

## Competing interests

The authors declare no competing interests
