## [Peer Review File · Nature Communications]

Recent upper Arctic Ocean warming expedited by summertime atmospheric processesReviewers' Comments:

Reviewer #1:

Remarks to the Author:

In this study, the authors consider the role of changing atmospheric conditions in driving upper ocean temperature trends in the Arctic over the past 40 years.

The work tackles an important and largely unanswered question in current polar climate science: how much of recent ocean warming in the Arctic is due to anthropogenic forcing and how much is due to internal climate variability? The authors aim to answer this question by performing nudging simulation where the anthropogenic climate forcing is held constant while the atmospheric circulation is tied to that of the observed winds. The results suggest up to a quarter of the observed 40-yr upper ocean warming is due to multi-decadal changes in wind conditions, with this ocean warming effected predominantly via a wind-induced reduction in upwelling shortwave radiation (due to changes in surface albedo from sea ice melt).

The study is well-conceived and carried out, and ties together a number of important processes of the Arctic climate system. I have a number of queries regarding the overall presentation, some details in the methodology, and also its place in the existing literature. Beyond these questions I will leave it to the editor to assess whether the findings presented here are of broad enough interest to warrant publication in Nature Communications.

1. Are the nudging experiments suited to isolate internal variability? The authors bring this up as a caveat on page 9. Since the observed wind changes may be strongly influenced by anthropogenic forcing, how can the conclusions of the nudging runs be phrased conclusively in terms of internal variability? Would it make sense to remove "internal" from the title? And since it appears that not a specific process is identified, maybe the title should read "... expedited by summertime atmospheric processes" ? Relatedly, I suggest rephrasing the numbers in the abstract and later on as bounds ("up to a quarter", and "up to 60%").
2. The authors put a lot of emphasis on the "lagged" response of the upper ocean to seasonal atmospheric warming. I am a little confused as to how this is different from the standard understanding that seasonal ocean warming is simply phase shifted due to its large heat capacity from that of the atmosphere (which has almost negligible heat capacity). In the limit of a deep mixed layer (and large heat reservoir) this phase lag is $\rightarrow \pi/2$, i.e. 3 months.
3. Similarly, the findings of atmospheric warming leading to upper ocean warming via reduced upwelling shortwave radiation are presented as novel and without much context of the existing literature.
4. The importance of variations in mixed layer depth, and in particular issues that GCMs have with accurately representing the strongly stratified upper ocean in the Arctic are not addressed in the manuscript.
5. It would be helpful to get some more details on how the nudging experiments are performed. For example, what is the relaxation timescale of the nudging? What are the differences in the initial conditions of the 5 ensemble members?
6. Beyond the time series of Fig 1a, little effort is spent on describing the upper ocean warming in the manuscript. In particular, discussing and quantifying the uncertainty due to the scarcity of observations would be helpful, as well as a more careful assessment of the regional differences.
7. The geopotential height at 300 hPa (Z300) features quite prominently throughout the manuscript, without much detail being offered on what an increase in Z300 means for the atmospheric circulation

patterns. Could the authors elaborate on this point?

8. In general, the language used is at points imprecise and somewhat difficult to follow. For example, on several occasions the authors talk about temperature and other "indices" but I believe they just refer to "temperatures"? Another example, the second sentence of the main text: 'a "new normal" of the Arctic with more sea ice-free summers and longer melting seasons by the middle of this century'. In a projected gradual decrease of sea ice extent, what is the "new normal"? What does "more sea ice-free summers" mean, since we haven't had any to date? And yes, longer melting seasons are expected by the middle of this century, but also have been observed in recent years and likely over the coming decades, and beyond. One more example: the choice of the term "pathways" is a little confusing (near the bottom of page 2). "processes" would be clearer, since "pathways" in this context prompted me to think of "pathways of water flux" or similar.

Reviewer #2:

Remarks to the Author:

This study investigates the effect of atmospheric internal variability on Arctic Ocean temperatures in recent decades. The manuscript convincingly links atmospheric warming in summer to upper-ocean warming in fall, before exploring the effect of internal atmospheric variability on ocean temperatures through nudging experiments. In explaining a quarter of the upper-ocean Arctic warming in the recent four decades, the manuscript highlights an important mechanism that seems to have been somewhat overlooked in the past. I therefore think the study will become a valuable contribution to our understanding of Arctic Ocean heat content.

The manuscript is well structured, the graphics are largely good, and it made for an enjoyable read. That being said, I want to urge the authors to be careful with referring to ocean reanalyses (here ORAS5) as 'observations', and saying that e.g. 'observed' (ORAS5) versus 'simulated' (CESM) ocean temperatures are compared. In my mind this is not accurate. Writing from an Atlantic perspective, I'm also missing some reflection and details with regards to the authors discussion of the Atlantic sector. See the more detailed comments below:

(Note that line numbers were not provided in the manuscript, which makes giving precise comments challenging).

Major comments:

1) Mechanisms linking JJA atmospheric changes with SON upper ocean temperatures:

The authors write that Fig. 1e 'feature a tilted downward intrusion starting from June-August at the surface and propagating downward toward 50m by fall', and describe it as a 'physical pathway linking upper ocean temperature in cold seasons in the Arctic with atmospheric warming in the summer'. This is a bit vague to me. Which mechanisms/processes are actually at play here? The mechanisms-subsection confirms the statistical link between Q_{net} and upper-ocean temperature, but doesn't really discuss any oceanic processes. For instance, I'm guessing a deepening of mixed layers from summer to winter (from brine rejection and wind-driven stirring from ice motion) is relevant for explaining the time lag at depth? To what depth does longwave and shortwave radiation penetrate if the ocean surface is ice free?

2) The Atlantic sector:

The Atlantic sector (APSS) is mentioned once (Page 5, paragraph 1) and not really discussed directly again. While the APSS falls outside the domain analyzed (Fig. 1b), it is clear from Fig. 1c that the

majority of the ocean warming is happening in the Atlantic sector. I therefore think some more discussion of the Atlantic sector is called for:

a) The 0-50m temperature trend in the nudging experiments (Fig. 3c) only seem to show regional warming close to the Bering Strait. Are the trends in the central Arctic Ocean even significant in Fig. 3c? If not, I think the authors need to specify (in abstract and conclusion) that the quarter of the pan-Arctic warming trend explained by the anomalous winds largely originate from a warming of the Chukchi and East Siberian seas.

b) From Fig. 4b-c it is clear that trends in 0-50m Arctic Ocean temperatures are much more related to the Atlantic inflow than the Pacific inflow. This makes me question the focus on the Pacific sector. A more thorough description of the regional differences and the relative importance would help.

c) Atlantification (page 12, paragraph 3): Mentioning Atlantification is certainly relevant here, but the sentence is not completely accurate. While ref. 26 and 50 (Polyakov et al., 2020 and Polyakov et al., 2017) focus on weakened stratification and increased vertical mixing in the Eurasian Basin, ref. 49 (Asbjornsen et al., 2020) focus on the Barents Sea where the governing processes are a little different. Ref. 49 ties Atlantification (warming, salinification, ice loss) in the Barents Sea to a warmer Barents Sea inflow and periods of reduced ocean heat loss, rather than increased vertical mixing.

3) Poleward ocean heat transport:

a) Heat transport is calculated relative to an arbitrary reference temperature (e.g. Schauer and Beszczynska-Moller, 2009; www.ocean-sci.net/5/487/2009/). I think the authors need to justify their choice of -1.8°C in the methods section. Is the POHT magnitude and variability in Fig. 4 sensitive to the choice of reference temperature? Traditionally, 0°C has been chosen for the Barents Sea Opening (e.g. ref. 19 Årthun et al., 2019 and references within) though this is also an arbitrary reference temperature.

b) Fig 4d-f and Page 11, paragraph 3: Yes, the Pacific gateway is a good match in terms of variability, but it's not really a good match in terms of magnitude (left-hand y-axis for CESM nudging runs and right-hand y-axis for ORAS5). Why?

c) Page 11, paragraph 3: Atlantic gate POHT in CESM historical simulation – 'None of these capture the observed POHT trend through the Atlantic Gate, suggesting that Atlantic POHT is independent from CO₂ and wind-driven forcing'. I find this hard to believe – see Fig. 2 in ref. 19 (Årthun et al., 2019; also using CESM 40 member ensemble). Increased POHT through the Barents Sea Opening is seen for both the historical period and in future projections (long-term trend driven by increasing inflow temperatures).

4) ORAS5 ocean reanalysis:

a) Page 7-13, Fig. 3a and Fig. 4 legends: I strongly disagree with the authors referring to the ocean reanalysis (ORAS5) as 'observations'. It is not common to refer to ocean reanalyses (ORAS5, SODA) or state estimates (ECCO, ASTE) as observations, as these are solutions for the ocean and sea-ice states produced by ocean-sea-ice coupled models driven by atmospheric surface forcing and constrained by ocean observations. Especially in the Arctic Ocean, observational constraints are few and one should be somewhat careful with interpreting the results. The only evaluation of ORAS5 presented is the comparison to UpTempO Buoy temperatures (Fig. 1a), which displays similar variability as the reanalysis products. I believe it is common practice to refer to atmospheric reanalyses as observations, and I will not object to this.

b) Methods section: The description of the ORAS5 observational constraints is too brief. Satellite sea level and sea ice concentration is mentioned, but what about subsurface constraints? Isn't Argo data used? Is data from Arctic Ice-Tethered profilers included? This is important information in terms of discussing how reliable the product is for the Arctic Ocean.

5) Page 9, paragraph 2: On the one hand the warming explained is likely an overestimate/upper bounds, on the other hand it's likely an underestimate? Do you have any idea about the magnitudes of these competing effects? While mentioning these caveats is very important, I would rephrase so that the authors don't directly contradict themselves.

6) Page 12, first sentence: the 'trend toward anomalous anticyclonic circulation over the Arctic Ocean and Greenland' is shown in the supplementary only. If this is to be highlighted as a key result, the authors should consider moving Fig. S1 or Fig. S3 to the main manuscript.

7) Fig. 1b-c, Fig. 2a-c, Fig3b-c, Fig. 4b-c: I recommend the authors improve their visualisation of regions with statistically significant trends and/or correlations. It's hardly visible in the figures (particularly Fig. 2a-c). In Fig. 4b-c, the black dots that I think is supposed to indicate significance looks very odd (spaced too far apart?). Is significance even indicated in Fig. 1c and Fig. 3b-c? I think it should be.

Minor comments:

1) Fig. 1b-c: Maps appear too small on page. Enlarging these subplots and reducing white space would help. Also, consider using a different color on the marked domain in b (black line on dark blue is not very visible).

2) Page 9, paragraph 1: The authors state that: "The unexplained portion is likely due to processes not captured by our nudging experiments and CESM-LEN". This is a bit vague – which processes are you thinking of exactly?

3) Page 10, paragraph 2, final line: '... reduces sea ice loss and upper ocean warming' – do you mean increases?

4) Page 10, second line of heat transport subsection: do you mean Fig. 3b&c?

5) Page 11, paragraph 2: Text says 'regressing the POHT time series with ocean temperature' while the caption of Fig. 4b-c says 'correlations'.

Reviewer #3:

Remarks to the Author:

Review of „ Recent upper Arctic Ocean warming expedited by a summertime internal atmospheric process “ by Li et al

This study explains the role of summertime internal atmospheric processes in the Arctic upper ocean warming. It suggests that a nonnegligible portion of the Arctic upper ocean warming in the past decades can be attributed to internal variability. The paper can be a significant contribution for improving our understanding of anthropogenically versus internally driven Arctic Ocean changes. The methods were well thought, and the major conclusion is supported by the analysis.

I have a few major comments.

1. The section "Poleward ocean heat transport contribution" and Figure 4 need major revision. First, there is no guarantee that only the heat inflow in the upper 50 m can influence the temperature in the upper 50 m, which is also mentioned in the discussion section. This is especially the case for the Atlantic water inflow. As no closed-budget analysis is planned, the authors can at least state that the variability of ocean heat transport from the Atlantic is largely the same for the upper 50 m and some depth below, if it is so. Second, the variability of Atlantic heat inflow through Barents Sea and Fram Strait is different, and these two branches could also have different impact on the ocean downstream. It is not a good idea to use the sum of the total heat transport in the analysis. Third, the correlation in Figure 4b,c is done without lag in time. It is hard for the Atlantic water heat to reach the Canadian Basin immediately, where the correlation is indicated to be significant in Figure 4b. The authors need to think how to better illustrate the impact of the ocean heat transport on the upper ocean, or provide mechanisms that support immediate long-distance impact.
2. The contribution of the internal variability is much larger after 2000. I would suggest to show related plots (currently Figure 3) for this period too, at least as supplementary material if it is hard to fit them into the main paper due to length limit. It is interesting to know whether the location of strong warming trends is the same in different periods.
3. Summing the ocean warming trends in the large ensemble mean and in the nudging runs produces a lower trend than in reanalysis. Please show these trends separately and their sum as well, as 2D plots (format like Figure 3b). One can see where the trend is underestimated in the simulations by comparing to Figure 3b. Is the underestimation located where the ocean heat transports can play an important role?

Some minor comments

1. Please provide line numbers. The pdf version I got has no line numbers, so it is hard to provide specific comments.
2. In figure 1d, the atmosphere warming trend is actually smaller in JJA than in other seasons. Need to write explicitly (when referring to this plot) that the sea ice feedback allows stronger impact on the ocean temperature, despite the air warming trend is not the largest in JJA.
3. Page 5, is it proper to put Laptev Sea in PPSS instead of APSS?
4. Page 6, between atmospheric warming and ocean, change to "atmospheric and ocean ..."
5. Page 7, "but this is a secondary effect from the reduction of multiple reflection resulting from a shrinkage of sea ice coverage." Although references are provided, it is still better to provide a few sentences explaining it directly inside this paper, as it is important to know the reason that you neglect this relatively large budget term.
6. Page 8, "the warming in the nudging simulations does not advance as far northward as in the observations." How about the recent 2 decades? See the major comments.

Page 9, "...are from a decades-long spin up which generates warmer states than the real condition in 1979. This means that our estimate of the role of internal variability could in fact be underestimated by a warm bias in the models' initial states..." How big is the artificial trend (numerical drift in temperature)? If you repeat the runs for 40 years, using exactly the same setup as the wind nudging runs, just without applying wind nudging, do you get warming or cooling trend in the upper Arctic Ocean? Without answering this question, the original sentence in the paper does not make much sense.

Page 10, capture over – capturing over

Page 10, in turn reduces – in turn increases?

Page 10, last paragraph, Figure 4 – Figure 3

Page 11, "This suggests that the wind-driven atmospheric circulation change plays an important role in driving POHT through the Bering Strait" This statement can be well supported by citing literature (Danielson2014, Zhang2020):

Danielson, S. L., Weingartner, T. J., Hedstrom, K. S., Aagaard, K., Woodgate, R., Curchitser, E., & Stabeno, P. J. (2014). Coupled wind forced controls of the Bering-Chukchi shelf circulation and the Bering Strait throughflow: Ekman transport, continental shelf waves, and variations of the Pacific-Arctic sea surface height gradient. *Progress in Oceanography*, 125, 40-61.

Zhang, W., Wang, Q., Wang, X., & Danilov, S. (2020). Mechanisms driving the interannual variability of the Bering Strait throughflow. *Journal of Geophysical Research: Oceans*, 125, e2019JC015308.

Page 11, "but that POHT changes from the Atlantic Ocean are likely determined by other factors, such as the initial ocean condition, deeper layer oceanic variability, internal oceanic thermohaline variability in the Arctic and heat transport from the lower latitudes where observed winds are not specified." The complication about explaining the variability of Atlantic water inflow can be well supported by citing related literature, for both Fram Strait and Barents Sea branches (e.g., Muilwijk2019, Wang2019, Wang2020):

Muilwijk, Morven, Ilicak, Mehmet, Cornish, Sam B., Danilov, Sergey, Gelderloos, Renske, Gerdes, Rüdiger, Haid, Verena, Haine, Thomas W. N., Johnson, Helen L., Kostov, Yavor, Kovács, Tamás, Lique, Camille, Marson, Juliana M., Myers, Paul G., Scott, Jeffery, Smedsrud, Lars H., Talandier, Claude and Wang, Qiang (2019) Arctic Ocean response to Greenland Sea wind anomalies in a suite of model simulations, *J. Geophys. Res.-Oceans*, 124, 6286-6322.

Wang, Q., Wang, X., Wekerle, C., Danilov, S., Jung, T., Koldunov, N., Lind, S., Sein, D., Shu, Q., Sidorenko, D. (2019). Ocean heat transport into the Barents Sea: Distinct controls on the upward trend and interannual variability, *Geophysical Research Letters*, 46, 13180-13190.

Wang, Q., Wekerle, C., Wang, X., Danilov, S., Koldunov, N., Sein, D., Sidorenko, D., von Appen, W. and Jung, T. (2020) Intensification of the Atlantic Water supply to the Arctic Ocean through Fram Strait induced by Arctic sea ice decline, *Geophysical Research Letters*, 47, e2019GL086682.

Page 2 where Arctic Ocean warming is mentioned, or Page 12-13, where Atlantification is mentioned, a recent very relevant paper is not cited:

Qi Shu, Qiang Wang, Zhenya Song, Fangli Qiao (2021). The poleward enhanced Arctic Ocean cooling machine in a warming climate. *Nature Communications*, 12, 2966.

Methods

Poleward ocean heat transport. Page 16, the transect of Atlantic Water inflow seems to be not a closed transect.

Figure 4 caption, "ten wind nudging experiments", is it ten?

All figures: figure quality needs to be better in the final version

Supplementary figure 5, the color range for temperature is different for the first and third row. Any reason?

The paper is largely well written, but proofreading is still needed.

We would like to extend our appreciation to all reviewers for their time to assess our work, their positive feedback and constructive comments on our manuscript. These comments and suggestions have helped to improve the presentation and clarity of our main points. We have incorporated the reviewers' suggestions into the revised manuscript, mainly including:

1. As suggested by the reviewers, we have modified the title to "Recent upper Arctic Ocean warming expedited by summertime atmospheric processes".
2. More analysis and relevant detailed discussions are provided to address reviewers' concerns regarding the robust mechanistic explanation for the identified link between atmospheric warming and subsequent ocean warming and its time lag.
3. A more thorough assessment of the quality of ORAS5 reanalysis is provided by comparing the dataset with in situ measurement and other reanalysis products.
4. More detailed discussion on differences between poleward ocean heat transports from the Pacific and Atlantic Gates are included.

Below are our detailed replies (in blue font) to each of their comments. We hope that we have sufficiently addressed all raised issues in the revised manuscript.

Reviewer #1 (Remarks to the Author):

In this study, the authors consider the role of changing atmospheric conditions in driving upper ocean temperature trends in the Arctic over the past 40 years.

The work tackles an important and largely unanswered question in current polar climate science: how much of recent ocean warming in the Arctic is due to anthropogenic forcing and how much is due to internal climate variability? The authors aim to answer this question by performing nudging simulation where the anthropogenic climate forcing is held constant while the atmospheric circulation is tied to that of the observed winds. The results suggest up to a quarter of the observed 40-yr upper ocean warming is due to multi-decadal changes in wind conditions, with this ocean warming effected predominantly via a wind-induced reduction in upwelling shortwave radiation (due to changes in surface albedo from sea ice melt).

The study is well-conceived and carried out, and ties together a number of important processes of the Arctic climate system. I have a number of queries regarding the overall presentation, some details in the methodology, and also its place in the existing literature. Beyond these questions I will leave it to the editor to assess whether the findings presented here are of broad enough interest to warrant publication in Nature Communications.

We thank the reviewer for the positive comments and the constructive suggestions. We agree with the overall comments and have revised our manuscript to address them accordingly.

1. Are the nudging experiments suited to isolate internal variability? The authors bring this up as a caveat on page 9. Since the observed wind changes may be strongly influenced by anthropogenic forcing, how can the conclusions of the nudging runs be phrased conclusively in terms of internal variability? Would it make sense to remove "internal" from the title? And since it appears that not a specific process is identified, maybe the title should read "... expedited by summertime atmospheric processes" ? Relatedly, I suggest rephrasing the numbers in the abstract and later on as bounds ("up to a quarter", and "up to 60%").

We agree with the reviewer that our original argument about the internal origin of observed wind changes can be improved.

To determine the role of anthropogenic warming on the high-latitude circulation, Ding et al. (2017) examined the ensemble average trends in JJA geopotential height and winds at 200 hPa over the Arctic region from the historical simulations of the Coupled Model Intercomparison Project Phase 5 (CMIP5) and the CESM-Large Ensemble (CESM-LEN). Since internal variability of simulations can be canceled out by doing ensemble average, the ensemble average of all members largely reflects changes due to anthropogenic forcing. Unlike ERA-I (Fig. R1a), which shows significant barotropic, anticyclonic wind trends over Greenland from the surface to the upper troposphere, the wind trends in CMIP5 and CESM-LEN ensembles (Fig. R1b&c) are insignificant and unorganized in the Arctic, suggesting that the observed wind trends are very unlikely driven by anthropogenic forcing. Furthermore, in this study, we use all 40 members from CESM-LEN to revisit this issue and find that this conclusion still holds very well for two periods from 1979 to 2018 and from 2000 to 2018 at a different pressure level. As shown in Fig. R2, JJA 300 hPa (upper tropospheric) wind trends over these two periods from the 40-member ensemble mean of CESM-LEN are very weak and at most account for a small part of observed trends in ERA5. This new comparison further confirms that the observed trends of JJA winds from the two periods are most likely due to internal climate variability rather than anthropogenic forcing. Since anthropogenic forcing is held constant in all ensemble members of our nudging runs, the results of the nudging simulation can be interpreted as the response to imposed winds that is mostly internally driven.

However, we also agree that the focus of this study is to emphasize the physical mechanism responsible for recent upper Arctic Ocean warming rather than highlighting its internal origin and that the mechanism contains a chain of sub-processes. So the title is modified to “**Recent upper Arctic Ocean warming expedited by summertime atmospheric processes**”, as the reviewer suggested. The manuscript is also revised to reflect that those quantifications may represent the upper bound of the contribution due to wind changes.

Response Fig. 1. Linear trend of JJA Z200 (m per decade) and zonal and meridional winds from (a) ERA-I, (b) the 26-model ensemble mean from CMIP5 project, and (c) 30-member ensemble mean from CESM-Large Ensemble (CESM-LEN) project (Adapted from Figure 4 of Ding et al. 2017, Nature Climate Change).

Response Fig. 2. Linear trends of JJA 300hPa wind ($m s^{-1}$ per decade) from the ERA5 reanalysis for the period 1979 - 2018 in **a** and the period 2000 - 2018 in **c** and from the 40-member ensemble mean of CESM-LEN for the period 1979 - 2018 in **b** and the period 2000 -2018 in **d**. (Supplementary Fig. 3 in the manuscript)

2. The authors put a lot of emphasis on the "lagged" response of the upper ocean to seasonal atmospheric warming. I am a little confused as to how this is different from the standard understanding that seasonal ocean warming is simply phase shifted due to its large heat capacity from that of the atmosphere (which has almost negligible heat capacity). In the limit of a deep mixed layer (and large heat reservoir) this phase lag is $\rightarrow \pi/2$, i.e. 3 months.

The use of a "lagged" analysis is designed to show the chain of causality from the atmosphere to the ocean (rather than the opposite direction which has been invoked in other studies). The timing of the lagged response is of secondary interest to our study. Nevertheless, the lagged response of the upper ocean to seasonal atmospheric warming is in line with our common understanding of the time (normally 3 months) taken by the ocean to warm up after receiving extra heat from the atmosphere due to ocean's large heat capacity, and it also takes time to melt sea ice. To make this point clearer, we add a few sentences to clarify it in lines 97-100 as below:

"It is clear that JJA atmospheric warming significantly precedes upper ocean warming from early summer to the following fall and even winter since it takes time to melt sea ice and then warm the ocean due to larger heat capacity of ocean and sea ice. This calculation suggests that atmospheric forcing drives ocean warming rather than the reverse in summer."

3. Similarly, the findings of atmospheric warming leading to upper ocean warming via

reduced upwelling shortwave radiation are presented as novel and without much context of the existing literature.

In the revision, we cited more literature and explained this finding in the context of the original literature in lines 161-162 as below:

“As the surface albedo decreases with sea ice melt, more solar radiation is absorbed by the darker ocean^{35,40,41}.”

35. Huang, Y., Ding, Q., Dong, X., Xi, B. & Baxter, I. Summertime low clouds mediate the impact of the large-scale circulation on Arctic sea ice. *Communications Earth & Environment* **2**, 1–10 (2021).

40. Serreze, M. C., Barrett, A. P., Stroeve, J. C., Kindig, D. N. & Holland, M. M. The emergence of surface-based Arctic amplification. *The Cryosphere* **3**, 11–19 (2009).

41. Serreze, M. C. & Barry, R. G. Processes and impacts of Arctic amplification: A research synthesis. *Global and Planetary Change* **77**, 85–96 (2011).

We also emphasize that the upwelling shortwave radiation is largely reduced as a result of decreased surface albedo which is caused by wind-induced warming in this study.

4. The importance of variations in mixed layer depth, and in particular issues that GCMs have with accurately representing the strongly stratified upper ocean in the Arctic are not addressed in the manuscript.

Thanks for pointing this out. We only briefly discussed this issue in our original manuscript by emphasizing the skill of the model in replicating some key features of the mean states of temperature and salinity along the vertical cross section Alaska-North Pole-Svalbard. To examine the model and reanalysis skill in representing the observed mixed layer depth (MLD), we calculated the climatology of MLD in JJA and SON from ORAS5, WOA18, and our wind nudging runs (Fig. R3). Summer and fall MLD in ORAS5 is shallower than observed in WOA18 and simulated one (Fig. R3). The magnitude of simulated MLD is closer to observed, but their patterns show some differences. It is known that most reanalyses and models have biases in replicating observed MLD and how MLD changed over the past decades remain as an open question due to large uncertainty of MLD observations (Uotila et al., 2019). Thus, in this study, we don't focus on the role of ocean mixing in regulating upper ocean temperature but just briefly discuss its importance at the end of the paper (lines 327-335). Alternatively, we find that most of the increasing trends in upper 50 m ocean temperature in our nudging simulations are forced by enhanced solar shortwave heat flux absorbed in the ocean boundary layer (Q_{short_bl}) (Fig. R4a). This calculation suggests that JJA Q_{net} at the surface leads to substantially more absorption of Q_{short_bl} in the following season within the Arctic (Fig. R4b&c). Therefore, we added more discussion about Q_{short_bl} in the revision in lines 246-259 to make the mechanisms clearer.

Response Fig. 3. a. Climatological (m) of mixed layer depth in JJA from ORAS5 in a, from WOA18 in b, and from the ensemble average of five wind nudging experiments in c for the period 2005 - 2017. **d-f.** The same as a-c, but for mean mixed layer depth in SON.

Response Fig. 4. a. Linear trend (Watt/m^2 per decade) of solar shortwave heat flux in the ocean boundary layer (Q_{short_bl}) field in SON from the ensemble average of five wind nudging experiments for the period 1979 – 2018. **b-c.** Correlation of JJA domain-average net heat flux (Q_{net}) with Q_{short_bl} field in JJA in **b**, and in SON in **c** from the ensemble average of five wind nudging experiments for the period 1979 – 2018. The domain used for the calculation is the area enclosed by the solid black contour in **Fig. 1b**. Black stippling in all plots indicates statistically significant correlations or trends at the 95% confidence level. (Supplementary Fig. 7 in the manuscript)

5. It would be helpful to get some more details on how the nudging experiments are performed. For example, what is the relaxation timescale of the nudging? What are the differences in the initial conditions of the 5 ensemble members?

We Agree. In short, we first have a long spin-up perpetual run (continuously forced by reanalysis winds of year 1979 in the Arctic) for 150 years to ensure that there will be no significant artificial drift after we impose the reanalysis winds in the following nudging runs. After almost 100 years, the simulated upper ocean temperature in the Arctic appears to be stabilized (Fig. R5). The initial conditions of our 5 nudging runs are then derived from Jan. 1 of the last 5 years of this long spin-up. In these five runs, we start to allow nudged winds to vary from 1979 to 2018 within the Arctic. The relaxation time scale of the nudging is around 6 hours since the nudging scheme operates in each model step. We provided more details on how we performed the wind nudging experiments in the revision in lines 405-425.

Response Fig. 5. The Arctic Ocean domain-average upper (0 – 50 m average) ocean temperature (°C) in SON during 150-yr spin-up perpetual run. The domain used for the calculation here is the area enclosed by the solid black contour in **Fig. 1b**. (Supplementary Fig. 11 in the manuscript)

6. Beyond the time series of Fig 1a, little effort is spent on describing the upper ocean warming in the manuscript. In particular, discussing and quantifying the uncertainty due to the scarcity of observations would be helpful, as well as a more careful assessment of the regional differences.

We agree. Even after a number of field expeditions and improved satellite coverage, the Arctic Ocean observations remain sparse compared to the North Atlantic (Uotila et al., 2019). To give readers a better sense of the quality of reanalyses, we calculated the upper ocean temperature trend for three reanalysis data and for WOA18, and the temperature difference between 1985 – 1994 and 2005 – 2017 since WOA18 only provides monthly decadal mean temperature fields for 1985-1994, 1995-2004, and 2005-2017 (Fig. R6&7). We also calculated the spatial correlation of temperature difference (Fig. R7) within the Arctic (north of 70°N) between WOA18 and those from three reanalyses (ORAS5: 0.74; SODA3.4.2: 0.65; GECCO3: 0.64) and found that ORAS5 has the highest spatial correlation with WOA18. We also calculated domain-average upper ocean temperature for three different decades to compare these with reanalyses in Fig. 1a. It is clear from these comparisons that the ORAS5 reanalysis has a reasonable skill to well capture changes that are reflected by WOA18 and UpTempO. In particular, its spatial pattern is consistent with that in WOA18 (Fig. R7). A more thorough assessment of ORAS5 with different reanalyses and observation is added in the revision in lines 356-377.

Response Fig. 6. a-c. Linear trend ($^{\circ}\text{C}$ per decade) of SON upper ocean temperature using three different reanalysis data (ORAS5 in **a**, SODA3.4.2 in **b**, and GECCO3 in **c**) (1979 - 2018). SODA3.4.2 only provides data from 1980 to 2016, which is different from the other two reanalyses. Black stippling in all plots indicates statistically significant trends at the 95% confidence level. (Supplementary Fig. 9 in the manuscript)

Response Fig. 7. a-d. Differences of SON upper ocean temperature between the periods 1985 – 1994 mean and 2005 – 2017 mean using three different reanalyses (ORAS5 in **a**, SODA3.4.2 in **b**, and GECCO3 in **c**) and observation data (WOA18 in **d**). SODA3.4.2 only provides data from 1980 to 2016, which excludes the data of year 2017 to calculate temperature difference. (Supplementary Fig. 10 in the manuscript)

7. The geopotential height at 300 hPa (Z300) features quite prominently throughout the manuscript, without much detail being offered on what an increase in Z300 means for the atmospheric circulation patterns. Could the authors elaborate on this point?

Geopotential height at a certain pressure level indicates the depth or thickness of an air column from the surface to the same pressure level, which usually reflects a warming of the whole air column between the surface and pressure level. In our case, we suggest that Z300 is changing towards a stronger anticyclonic circulation over Greenland and the Arctic Ocean (increases in Z300) over the past four decades. This means that the air column from the surface to the pressure level at 300 hpa became warmer over the period. However, this warming is not necessary generated by thermodynamic processes such as radiative forcing (e.g. greenhouse gases). Dynamical processes (wind driven) can also induce such warming through adiabatic subsidence. We synthesized elements of previous research on this topic into the PARC mode concept in the paper by Baxter et al. (2020) which is simply summarized in Fig. R8. Due to surface friction in the boundary layer of the atmosphere, anticyclonic winds (back thin arrows in the Fig. R8) at the surface tend to diverge outward and then lead to downward movement of air in the troposphere. This subsidence adiabatically warms the lower atmosphere (Fig. R8) and increases the geopotential height at boundary of upper troposphere, such as Z300. We added one sentence to explain the Z300 when we first mention it in the revision in lines 107-108 as below:

“Z300: its variability is a measure of temperature variations of the entire air column below 300 hPa”

The summer PARC mechanism

Response Fig. 8. A schematic diagram summarizing the formation mechanism of the high pressure over Greenland and the Arctic Ocean (Adapted from Figure 13 of Baxter et al., 2020).

8. In general, the language used is at points imprecise and somewhat difficult to follow. For example, on several occasions the authors talk about temperature and other "indices" but I believe they just refer to "temperatures"? Another example, the second sentence of the main text: 'a "new normal" of the Arctic with more sea ice-free summers and longer melting seasons by the middle of this century'. In a projected gradual decrease of sea ice extent, what is the "new normal"? What does "more sea ice-free summers" mean, since we haven't had any to date? And yes, longer melting seasons are expected by the middle of this century, but also have been observed in recent years and likely over the coming decades, and beyond. One more example: the choice of the term "pathways" is a little confusing (near the bottom of page 2). "processes" would be clearer, since "pathways" in this context prompted me to think of "pathways of water flux" or similar.

Thanks for drawing our attention on these minor issues. As suggested, we modified all statements about "index/indices" to be clearer and consistent. We corrected "pathways" to "processes". We also removed the sentence containing "new normal" to avoid confusion. We have thoroughly revised the manuscript to clean up all problems that could potentially confuse readers.

References:

- Baxter, I. *et al.* How Tropical Pacific Surface Cooling Contributed to Accelerated Sea Ice Melt from 2007 to 2012 as Ice Is Thinned by Anthropogenic Forcing. *Journal of Climate* **32**, 8583–8602 (2019).
- Ding, Q. *et al.* Influence of high-latitude atmospheric circulation changes on summertime Arctic sea ice. *Nature Climate Change* **7**, 289–295 (2017).
- Uotila, P. *et al.* An assessment of ten ocean reanalyses in the polar regions. *Clim Dyn* **52**, 1613–1650 (2019).

Reviewer #2 (Remarks to the Author):

This study investigates the effect of atmospheric internal variability on Arctic Ocean temperatures in recent decades. The manuscript convincingly links atmospheric warming in summer to upper-ocean warming in fall, before exploring the effect of internal atmospheric variability on ocean temperatures through nudging experiments. In explaining a quarter of the upper-ocean Arctic warming in the recent four decades, the manuscript highlights an important mechanism that seems to have been somewhat overlooked in the past. I therefore think the study will become a valuable contribution to our understanding of Arctic Ocean heat content.

We appreciate the positive comments on our work and also thank the reviewer for the constructive comments, especially the one suggesting us to clarify the mechanism linking the atmosphere and ocean. We have revised our manuscript to address all raised concerns.

The manuscript is well structured, the graphics are largely good, and it made for an enjoyable read. That being said, I want to urge the authors to be careful with referring to ocean reanalyses (here ORAS5) as ‘observations’, and saying that e.g. ‘observed’ (ORAS5) versus ‘simulated’ (CESM) ocean temperatures are compared. In my mind this is not accurate. Writing from an Atlantic perspective, I'm also missing some reflection and details with regards to the authors discussion of the Atlantic sector. See the more detailed comments below:

We fully agree that ORAS5 cannot be referred as “observation”. We rewrote all parts to refer ORAS5 as “reanalysis” in the revised version of the manuscript. We also devote more effort in the revision to discuss the POHT and ocean temperature changes over the Atlantic sector. Please see our detailed response below to your corresponding concerns in that regard.

(Note that line numbers were not provided in the manuscript, which makes giving precise comments challenging).

Done.

Major comments:

1) Mechanisms linking JJA atmospheric changes with SON upper ocean temperatures:

The authors write that Fig. 1e ‘feature a tilted downward intrusion starting from June-August at the surface and propagating downward toward 50m by fall’, and describe it as a ‘physical pathway linking upper ocean temperature in cold seasons in the Arctic with atmospheric warming in the summer’. This is a bit vague to me. Which mechanisms/processes are actually at play here? The mechanisms-subsection confirms the statistical link between Qnet and upper-ocean temperature, but doesn’t really discuss any oceanic processes. For instance, I’m guessing a deepening of mixed layers from summer to winter (from brine rejection and wind-driven stirring from ice motion) is relevant for explaining the time lag at depth? To what depth does longwave and shortwave radiation penetrate if the ocean surface is ice free?

Thanks for drawing our attention to this issue. To better understand the oceanic processes that link heat fluxes from the atmosphere with upper ocean temperature, we examine the solar shortwave heat flux into the ocean boundary layer (Q_{short_bl}).

Since longwave radiation doesn't penetrate, we focus on the penetration of solar shortwave into the ocean boundary layer in the wind nudging experiments, and trace its linkage to Q_{net} at the ice-atmosphere interface to explain the time lag. The shortwave radiation can directly penetrate into the boundary layer of the ocean which is ~ 30 m deep in the Arctic in wind nudging experiments (Fig. R9a). As shown in Fig. R10a, The increasing trends of Q_{short_bl} in SON are primarily confined to the Pacific sector of the Arctic Ocean. And Q_{net} absorbed at the surface in summer can lead to higher Q_{short_bl} in the central Arctic Ocean in fall (Fig. R10b&c). In the meantime, the ocean mixing homogenizes the solar shortwave heat flux in the boundary layer with deepening of mixed layer from summer to winter (Sirevaag et al., 2011; Steele et al., 2011). We added Fig. 3b to reveal this linkage better with the time lag in the revision. Since we primarily focus on thermodynamical processes linking the JJA atmospheric warming with the SON ocean temperature rise, we don't analyze the role of ocean mixing associated with this atmosphere-ocean linkage since reanalyses still contain large uncertainties to reflect the mean state and changes of mixed layer depth in the Arctic (please see our response to Reviewer 1's Point 4 regarding a similar concern, page 5). While our approach lacks a very careful examination of the role of oceanic mixing in warming the upper ocean temperature, we don't believe this limitation would affect the fundamental argument of the causal chain for ocean warming established in this paper: Subsidence and adiabatic warming from the atmosphere in summertime warms the atmosphere, melts sea ice, then the resulting open ocean can absorb more shortwave radiation. Thus, we added more detailed explanation on the mechanism discussed above in the revision in lines 246-259.

Response Fig. 9. **a.** Climatology (m) of boundary layer depth in SON from the ensemble average of five wind nudging experiments for the period 1979 – 2018. **b-c.** Linear trend (m per decade) of mixed layer depth in SON for the period 1979 – 2018 in **b**, and for the period 2000 - 2018 in **c** from the ensemble average of five wind nudging experiments. Black stippling in all plots indicates statistically significant trends at the 95% confidence level.

Response Fig. 10. **a.** Linear trend (Watt/m^2 per decade) of solar shortwave heat flux in the ocean boundary layer (Q_{short_bl}) field in SON from the ensemble average of five wind nudging experiments for the period 1979 – 2018. **b-c.** Correlation of JJA domain-average net heat flux (Q_{net}) with Q_{short_bl} field in JJA in **b**, and in SON in **c** from the ensemble average of five wind nudging experiments for the period 1979 – 2018. The domain used for the calculation is the area enclosed by the solid black contour in **Fig. 1b**. Black stippling in all plots indicates statistically significant correlations or trends at the 95% confidence level. (Supplementary Fig. 7 in the manuscript)

2) The Atlantic sector:

The Atlantic sector (APSS) is mentioned once (Page 5, paragraph 1) and not really discussed directly again. While the APSS falls outside the domain analyzed (Fig. 1b), it is clear from Fig. 1c that the majority of the ocean warming is happening in the Atlantic sector. I therefore think some more discussion of the Atlantic sector is called for:

a) The 0-50m temperature trend in the nudging experiments (Fig. 3c) only seem to show regional warming close to the Bering Strait. Are the trends in the central Arctic Ocean even significant in Fig. 3c? If not, I think the authors need to specify (in abstract and conclusion) that the quarter of the pan-Arctic warming trend explained by the anomalous winds largely originate from a warming of the Chukchi and East Siberian seas.

As suggested, we added a significance test for upper ocean temperature trends for the periods 1979 – 2018 and 2000 - 2018 (Fig. 3c-f). As shown in Fig. 3d, the trends in the central Arctic Ocean (1979 - 2018) in the nudging runs are not significant. However, the patterns of trends of the upper ocean temperature for the 2000 - 2018 period from nudging experiments look quite similar to that in ORAS5 (Fig. 3e&f; the spatial correlation within the Arctic (north of 70°N) reaches 0.77), and they are significant in both PPSS (especially over the Beaufort and Chukchi Seas, highlighted by dots in Fig. 3e&f) and APSS (especially over the Laptev, Barents, and Greenland Seas, highlighted in Fig. 3e&f). To be more specific, we rewrote that part to emphasize the regional differences of upper warming for the two periods (1979 - 2018 vs. 2000 - 2018) in lines 199-204 as below:

“The nudging experiment reproduces the spatial pattern of upper ocean temperature trends in ORAS5 in the PPSS from 1979 to 2018 and in the PPSS and APSS for the 2000 - 2018 period, although the temperature increases are slightly weaker (Fig. 3c-f). It is particularly

noted that the warming in the Barents Sea in the nudging simulations bears strong resemblance to ORAS5 for the 2000-2018 period (Fig. 3e&f), with the spatial correlation coefficient between these two trend patterns (Fig. 3e&f) within the Arctic (north of 70°N) reaching 0.77.”

We also mentioned in main text and conclusion that the upper ocean warming explained by the winds in our nudging runs for the past 40 years is largely confined within the Chukchi, East Siberian, and Laptev Seas.

b) From Fig. 4b-c it is clear that trends in 0-50m Arctic Ocean temperatures are much more related to the Atlantic inflow than the Pacific inflow. This makes me question the focus on the Pacific sector. A more thorough description of the regional differences and the relative importance would help.

We agree that our nudging runs do well over the Pacific sector of the basin and less successfully over the Atlantic sector, although POHT through the Atlantic Gate is more related with upper ocean temperature. We rewrote the paper to emphasize that the mechanism we raised here mainly operates over the Pacific sector of the Arctic Ocean for the past 40 years, but it operates well over the whole basin for the 2000-2018 period. We also added more discussion on the regional differences and relative importance of POHT from the two gates in the revision in lines 277-281 and 282-288 as below:

“Correlating the POHT time series with upper ocean temperature field shows that the variability of POHT through the Atlantic Gate in SON strongly affects the Barents and Kara Seas, and parts of the central Arctic Ocean (Fig. 4b). In contrast, POHT through the Pacific Gate has an increasing trend as well (Fig. 4a) but only affects the Chukchi Sea (Fig. 4c).”

“The simulated SON upper 50 m POHT through the Pacific Gate in the wind nudging runs very successfully capture the counterpart in ORAS5 (Fig. 4e; $r = 0.9$ with trends, $r = 0.89$ without trends), but this is not the case for the Atlantic Gate (Fig. 4d; Supplementary Fig. 8). This suggests that the wind-driven atmospheric circulation change plays an important role in driving POHT through the Bering Strait^{50,51}. Nevertheless, wind-driven POHT through the Bering Strait only has a weak correlation with upper ocean temperature in the interior of the basin and appears to have little impact on Pan-Arctic Ocean warming.”

c) Atlantification (page 12, paragraph 3): Mentioning Atlantification is certainly relevant here, but the sentence is not completely accurate. While ref. 26 and 50 (Polyakov et al., 2020 and Polyakov et al., 2017) focus on weakened stratification and increased vertical mixing in the Eurasian Basin, ref. 49 (Asbjornsen et al., 2020) focus on the Barents Sea where the governing processes are a little different. Ref. 49 ties Atlantification (warming, salinification, ice loss) in the Barents Sea to a warmer Barents Sea inflow and periods of reduced ocean heat loss, rather than increased vertical mixing.

Thanks for the correction. We removed Ref. 49 to ensure that we cite the references appropriately.

3) Poleward ocean heat transport:

a) Heat transport is calculated relative to an arbitrary reference temperature (e.g. Schauer and Beszczynska-Moller, 2009; www.ocean-sci.net/5/487/2009/). I think the authors need to

justify their choice of -1.8°C in the methods section. Is the POHT magnitude and variability in Fig. 4 sensitive to the choice of reference temperature? Traditionally, 0°C has been chosen for the Barents Sea Opening (e.g. ref. 19 Årthun et al., 2019 and references within) though this is also an arbitrary reference temperature.

Thanks for pointing this out. -1.9°C has been widely used in the calculations about heat fluxes through the Bering Strait (Woodgate, 2018; Woodgate et al., 2015; Woodgate et al., 2012; Woodgate et al., 2010). To be consistent with past literatures, we changed the reference temperature to -1.9°C to recalculate POHT through the Pacific Gate. We also agree that we should choose 0°C for the Atlantic Gate as you suggested.

The absolute values of heat fluxes are highly dependent on the reference temperature, but we think anomalies is not dependent on that. We updated all calculations related to PHOT in the revision.

b) Fig 4d-f and Page 11, paragraph 3: Yes, the Pacific gateway is a good match in terms of variability, but it's not really a good match in terms of magnitude (left-hand y-axis for CESM nudging runs and right-hand y-axis for ORAS5). Why?

We don't fully understand this mismatch either. One possible source of this mismatch could be due to mean state bias in the model, which is probably more noticeable over the Atlantic sector since many factors (e.g. anthropogenic forcing, the AMOC) are either not considered or not well constrained in our nudging runs. So we think seeing this discrepancy over the Atlantic sector is understandable. Definitely, more efforts should be devoted to further understand the underlying cause of this mismatch in future studies. We have added this discussion in the caption of Fig. 4 in the revision. In addition, a good match of variability over the Pacific sector gives us more confidence about the mechanism proposed in this study since we find that this mechanism's impacts are more prominent over the Pacific sector. We have emphasized this point in the revision.

c) Page 11, paragraph 3: Atlantic gate POHT in CESM historical simulation – ‘None of these capture the observed POHT trend through the Atlantic Gate, suggesting that Atlantic POHT is independent from CO_2 and wind-driven forcing’. I find this hard to believe – see Fig. 2 in ref. 19 (Årthun et al., 2019; also using CESM 40 member ensemble). Increased POHT through the Barents Sea Opening is seen for both the historical period and in future projections (long-term trend driven by increasing inflow temperatures).

We agree with you that this sentence is misleading. We updated Fig. 4d&e by adding SON POHT calculated from CESM-LEN 40 members. It appears that the trends of SON upper 50 m POHT through the Atlantic Gate driven by anthropogenic forcing is very weak compared with the one in ORAS5, indicating that both CO_2 and winds in our model cannot fully explain the observed upper 50 m SON POHT change (the Atlantic sector) in ORAS5. This weak trends of SON POHT through the Atlantic Gate in CESM-LEN can be probably explained by a decreasing trend of upper 50 m poleward volume transport in the same season during the period, which can offset the contribution from CO_2 caused ocean warming. The discrepancy between this result with Arthun et al. (2019) is probably because Arthun et al. (2019) focused on different seasons (annual average), location and depth. However, both our calculation and Arthun et al. (2019) found a slight slow-down of volume transport in and around the region. All these add uncertainties to our understanding of impacts of CO_2 forcing in driving POHT. We therefor believe this issue remains an open question. We modified this sentence in the

revision in lines 290-298 as below because the original statement didn't reflect our thought very well.

“None of these capture the increasing trend of POHT through the Atlantic Gate as seen in ORAS5, suggesting that SON upper 50 m POHT changes through the Atlantic Gate are likely determined by more complex factors that are not directly driven by winds and anthropogenic forcing in our model, such as the initial ocean condition, deeper layer oceanic variability, internal oceanic thermohaline variability in the Arctic and heat transport from the lower latitudes where observed winds are not specified^{19,52-54}. The role of anthropogenic forcing in contributing to increasing POHT via the Atlantic sector remains an open question since this attribution appears to be sensitive to approaches used to detect this feature^{19,55}.”

Response Fig. 11. Poleward volume transport (Sv) through the Atlantic Gate (the same as that used for the calculation of POHT) in the upper 50 m in SON from CESM-LEN 40 members (grey lines, and ensemble average in black line).

4) ORAS5 ocean reanalysis:

a) Page 7-13, Fig. 3a and Fig. 4 legends: I strongly disagree with the authors referring to the ocean reanalysis (ORAS5) as ‘observations’. It is not common to refer to ocean reanalyses (ORAS5, SODA) or state estimates (ECCO, ASTE) as observations, as these are solutions for the ocean and sea-ice states produced by ocean–sea-ice coupled models driven by atmospheric surface forcing and constrained by ocean observations. Especially in the Arctic Ocean, observational constraints are few and one should be somewhat careful with interpreting the results. The only evaluation of ORAS5 presented is the comparison to UpTempO Buoy temperatures (Fig. 1a), which displays similar variability as the reanalysis products. I believe it is common practice to refer to atmospheric reanalyses as observations, and I will not object to this.

We agree with you. We rewrote all parts to refer ORAS5 as “reanalysis” in the revised version of the manuscript. As also stated in our response to reviewer #1, we provided more careful assessment of ORAS5 with other reanalysis products and observations in the revision in lines 356-377.

b) Methods section: The description of the ORAS5 observational constraints is too brief.

Satellite sea level and sea ice concentration is mentioned, but what about subsurface constraints? Isn't Argo data used? Is data from Arctic Ice-Tethered profilers included? This is important information in terms of discussing how reliable the product is for the Arctic Ocean.

Done. In ORAS5, sea surface temperature is constrained to observational data, and subsurface ocean temperature/salinity and in situ temperature/salinity profiles are assimilated using 3DVar-FGAT procedure. The length of the data assimilation window is 5 days. As we know, sea ice data in ORAS5 partially relies on the dynamic-thermodynamic sea ice model LIM2, and the data from the Arctic Ice-Tethered profilers is not included. We added more information about ORAS5 reanalysis assimilation in the revision in lines 345-355.

5) Page 9, paragraph 2: On the one hand the warming explained is likely an overestimate/upper bounds, on the other hand it's likely an underestimate? Do you have any idea about the magnitudes of these competing effects? While mentioning these caveats is very important, I would rephrase so that the authors don't directly contradict themselves.

We agree. We think it is still premature to argue that a warming bias in the initial condition may have a significant impact in estimating the following trend driven by imposed winds. So we removed that sentence to avoid a possible contradiction.

6) Page 12, first sentence: the 'trend toward anomalous anticyclonic circulation over the Arctic Ocean and Greenland' is shown in the supplementary only. If this is to be highlighted as a key result, the authors should consider moving Fig. S1 or Fig. S3 to the main manuscript.

Done. We moved the original Fig. S1 to the new Fig. 1d. Thanks for your suggestion.

7) Fig. 1b-c, Fig. 2a-c, Fig3b-c, Fig. 4b-c: I recommend the authors improve their visualisation of regions with statistically significant trends and/or correlations. It's hardly visible in the figures (particularly Fig. 2a-c). In Fig. 4b-c, the black dots that I think is supposed to indicate significance looks very odd (spaced too far apart?). Is significance even indicated in Fig. 1c and Fig. 3b-c? I think it should be.

Done. We have adjusted all significance dots in new figures for better visualization, and indicated the significance for Fig. 1c & Fig. 3b-c.

Minor comments:

1) Fig. 1b-c: Maps appear too small on page. Enlarging these subplots and reducing white space would help. Also, consider using a different color on the marked domain in b (black line on dark blue is not very visible).

We have adjusted Figure 1 with larger subplots, and we have changed the color of the grid line to gray to highlight the elements in the subplots. We also adjusted the width of the marked domain contour for better visibility.

2) Page 9, paragraph 1: The authors state that: "The unexplained portion is likely due to processes not captured by our nudging experiments and CESM-LEN". This is a bit vague – which processes are you thinking of exactly?

We think the cause of this discrepancy remains as an open question. Our hunch is that the unexplained portion could be due to internal oceanic processes which are not captured by nudging experiments and ensemble average of CESM-LEN. Since the original sentence doesn't provide a useful information, we deleted it in the revision.

3) Page 10, paragraph 2, final line: '... reduces sea ice loss and upper ocean warming' – do you mean increases?

Corrected. "reduces" should be "increases".

4) Page 10, second line of heat transport subsection: do you mean Fig. 3b&c?

Corrected. It indicates Fig. 3b&c.

5) Page 11, paragraph 2: Text says 'regressing the POHT time series with ocean temperature' while the caption of Fig. 4b-c says 'correlations'.

Corrected. "regressing" should be "correlating".

References:

Sirevaag, A. et al. Mixing, heat fluxes and heat content evolution of the Arctic Ocean mixed layer. *Ocean Sci.* **7**, 335–349 (2011).

Steele, M., Ermold, W. & Zhang, J. Modeling the formation and fate of the near-surface temperature maximum in the Canadian Basin of the Arctic Ocean. *Journal of Geophysical Research: Oceans* **116**, (2011).

Woodgate, R. A. Increases in the Pacific inflow to the Arctic from 1990 to 2015, and insights into seasonal trends and driving mechanisms from year-round Bering Strait mooring data. *Progress in Oceanography* **160**, 124–154 (2018).

Woodgate, R. A., Weingartner, T. & Lindsay, R. The 2007 Bering Strait oceanic heat flux and anomalous Arctic sea-ice retreat. *Geophysical Research Letters* **37**, (2010).

Woodgate, R. A., Weingartner, T. J. & Lindsay, R. Observed increases in Bering Strait oceanic fluxes from the Pacific to the Arctic from 2001 to 2011 and their impacts on the Arctic Ocean water column. *Geophysical Research Letters* **39**, (2012).

Woodgate, R., Stafford, K. & Prahl, F. A Synthesis of Year-Round Interdisciplinary Mooring Measurements in the Bering Strait (1990–2014) and the RUSALCA Years (2004–2011). *Oceanog* **28**, 46–67 (2015).

Reviewer #3 (Remarks to the Author):

Review of „Recent upper Arctic Ocean warming expedited by a summertime internal atmospheric process “ by Li et al

This study explains the role of summertime internal atmospheric processes in the Arctic upper ocean warming. It suggests that a nonnegligible portion of the Arctic upper ocean warming in the past decades can be attributed to internal variability. The paper can be a significant contribution for improving our understanding of anthropogenically versus internally driven Arctic Ocean changes. The methods were well thought, and the major conclusion is supported by the analysis.

We appreciate that you spoke highly of our work. We also thank the reviewer for the constructive comments regarding ocean heat transport and wind nudging experiments. We have revised our manuscript to address the reviewer’s concerns accordingly, as clarified in greater detail below.

I have a few major comments.

1. The section “Poleward ocean heat transport contribution” and Figure 4 need major revision. First, there is no guarantee that only the heat inflow in the upper 50 m can influence the temperature in the upper 50 m, which is also mentioned in the discussion section. This is especially the case for the Atlantic water inflow. As no closed-budget analysis is planned, the authors can at least state that the variability of ocean heat transport from the Atlantic is largely the same for the upper 50 m and some depth below, if it is so.

We agree with you that we should emphasize this point more explicitly. As shown in Fig. R12, upper 50 m ocean heat transport through the Atlantic is similar to that in upper 300/2500 m (the Barents Sea ~ 300 m deep; the Fram Strait ~ 2500 m) because the Atlantic water inflow is largely barotropic. Considering that upper 50 m ocean temperature is the most sensitive to the heat inflow in the upper 50 m, we primarily focus on the POHT at this level in this study. On the other hand, the heat inflow in the upper 50 m through the Bering Strait is sufficient to modify the ocean temperature in the upper 50 m because the Bering Strait is only 50 m deep. To be more clear, we discuss this issue in the revision in lines 324-325 and 464-469 as below:

“In particular, the barotropic feature of the POHT through the Atlantic Gate (see Methods) requires an integrated view of heat transport throughout the whole depth.”

“The heat inflow in the upper 50 m through the Bering Strait is sufficient to modify the ocean temperature in the upper 50 m because the Bering Strait is only 50 m deep. Over the Atlantic sector, the variability of POHT through the Atlantic Gate from 0 – 300 m or 0 – 2500 m is similar to that in the upper 50 m because the Atlantic water inflow is largely barotropic (Supplementary Fig. 12). Considering that upper 50 m ocean temperature is the most sensitive to the heat inflow in the upper 50 m, we primarily focus on the POHT at this level in this study.”

Response Fig. 12. SON POHT (TW) through the Atlantic Gate in the upper 50 m, 300 m, and 2500 m from 1979 to 2018 using the ORAS5 reanalysis. (Supplementary Fig. 12 in the manuscript)

Second, the variability of Atlantic heat inflow through Barents Sea and Fram Strait is different, and these two branches could also have different impact on the ocean downstream. It is not a good idea to use the sum of the total heat transport in the analysis.

Thanks for letting us know this concern, but we think our main conclusion for this section is not sensitive to the choice of these two branches or the sum of them.

We calculated the SON POHT through the Fram Strait and Barents Sea in ORAS5 and wind nudging experiments, and their correlations with upper ocean temperature in SON. The upward trend of POHT through the Atlantic Gate as we observed in ORAS5 is mainly via the branch of the Barents Sea (Fig. R13). These two branches have different impacts on the upper ocean temperature (Fig. R14). However, the variabilities of POHT through both these two branches are not captured by the wind nudging experiments (Fig. R13), which means that our claim (which is that our model cannot fully capture the variability of POHT through the Atlantic Gate) doesn't depend on using either one of these two branches or their combination. In addition, as we know, POHT via these two branches can be sometimes in phase or out of phase, and their interaction remains an open question. Thus, further detailed analysis about two branches of the Atlantic heat inflow is needed but is beyond the scope of this study. We also added the Response Figure 13 as Supplementary Figure 8 in the revision to show the main difference of the two branches in ORAS5 and our wind nudging runs.

Response Fig. 13. SON POHT (TW) within upper 50 m from 1979 to 2018 through the Fram Strait in **a**, and through the Barents Sea in **b** from the ensemble average of five wind nudging experiments and the ORAS5 reanalysis. (Supplementary Fig. 8 in the manuscript)

Response Fig. 14. a-b. Correlations of SON upper ocean temperature field with SON upper 50 m POHT through the Fram Strait in **a**, through the Barents Sea in **b**, from the ORAS5 reanalysis (1979 - 2018). All linear trends are removed in calculating the correlations. Black stippling in all plots indicates statistically significant correlations at the 95% confidence level.

Third, the correlation in Figure 4b,c is done without lag in time. It is hard for the Atlantic water heat to reach the Canadian Basin immediately, where the correlation is indicated to be significant in Figure 4b. The authors need to think how to better illustrate the impact of the ocean heat transport on the upper ocean, or provide mechanisms that support immediate long-distance impact.

We agree that ocean currents are very slow, compared to winds, and it is hard for the Atlantic heat to reach the Canadian Basin immediately. However, Zhang et al. (1998) found a similar zero-lag correlation of POHT through the Atlantic Gate and pan-Arctic ocean heat content. Zhang et al. (1998) found this correlation because (i) the area-averaged ocean heat content includes the area very close to the gates chosen for the POHT, so a significant correlation with a zero or short lag as shown by Zhang et al. (1998) may contain some auto-correlations, and (ii) there could be an air-sea heat flux exchange that quickly takes the warm signal from the Barents Sea via the atmosphere and communicates it to other areas of the Arctic Ocean. In addition, the Atlantic Gate we used in the study (along 76°N) is closer to the central Arctic Ocean, so it makes easier for the oceanic heat to be transported to the central ocean and the Canadian Basin. These possible reasons could explain the significant correlations shown in the central Arctic Ocean in Fig. 4b, although we believe a more comprehensive analysis is needed to understand the fundamental causes of the immediate long-distance relationship the reviewer pointed out.

2. The contribution of the internal variability is much larger after 2000. I would suggest to show related plots (currently Figure 3) for this period too, at least as supplementary material if it is hard to fit them into the main paper due to length limit. It is interesting to know whether the location of strong warming trends is the same in different periods.

Thanks for your suggestion. We have added two subplots (Fig. 3e&f) in the new Fig. 3 to show the upper ocean warming trends for the period 2000 - 2018 in ORAS5 and wind nudging experiments. As seen in Fig. 3e&f, the nudging experiments reproduced very well the spatial pattern of upper ocean temperature trends in ORAS5 over the whole basin for the period 2000 - 2018 (the spatial correlation within the Arctic (north of 70°N) reaches 0.77).

We added relevant discussion about spatial differences of upper ocean warming trends over the period 1979 to 2018 and recent 19 years 2000 - 2018 in the revision in lines 199-204 as below:

“The nudging experiment reproduces the spatial pattern of upper ocean temperature trends in ORAS5 in the PPSS from 1979 to 2018 and in the PPSS and APSS for the 2000 - 2018 period, although the temperature increases are slightly weaker (Fig. 3c-f). It is particularly noted that the warming in the Barents Sea in the nudging simulations bears strong resemblance to ORAS5 for the 2000-2018 period (Fig. 3e&f), with the spatial correlation coefficient between these two trend patterns (Fig. 3e&f) within the Arctic (north of 70°N) reaching 0.77.”

3. Summing the ocean warming trends in the large ensemble mean and in the nudging runs produces a lower trend than in reanalysis. Please show these trends separately and their sum as well, as 2D plots (format like Figure 3b). One can see where the trend is underestimated in the simulations by comparing to Figure 3b. Is the underestimation located where the ocean heat transports can play an important role?

As suggested, we calculated the combined warming trends from the both CESM-LEN and nudging runs. Fig. R15&16 show that the simulated warming (CO_2 + winds) around the Greenland and Barents Seas is slightly underestimated compared with that in ORAS5. And these include areas where the ocean heat transports through the Atlantic Gate can play an important role. This indicates that CESM may still have difficulties to fully capture oceanic variability over the Atlantic sector, no matter it is forced by anthropogenic or wind forcing. We emphasized in the revision that our mechanism operates better in the Pacific sector of the Arctic Ocean for the past 40 years, and the ocean warming explained by wind-driven adiabatic atmospheric warming is mainly confined within the Pacific sector.

Response Fig. 15. a-b & d. Linear trend ($^{\circ}\text{C}$ per decade) of upper (0 – 50 m average) ocean temperature from the ensemble average of five wind nudging experiments in **a**, and from the CESM-LEN 40 members average in **b**, and from the ORAS5 reanalysis in **d** for the period 1979 - 2018. **c.** Summing the linear trend ($^{\circ}\text{C}$ per decade) of upper ocean temperature from the ensemble average of five wind nudging experiments (a) and CESM-LEN 40 members average (b) for the period 1979 - 2018.

Response Fig. 16. Difference of upper ocean temperature linear trend ($^{\circ}\text{C}$ per decade) between the summing of that in the ensemble average of five wind nudging experiments with CESM-LEN 40 members average and that in the ORAS5 reanalysis for the period 1979 - 2018.

Some minor comments

1. Please provide line numbers. The pdf version I got has no line numbers, so it is hard to provide specific comments.

Added.

2. In figure 1d, the atmosphere warming trend is actually smaller in JJA than in other seasons. Need to write explicitly (when referring to this plot) that the sea ice feedback allows stronger impact on the ocean temperature, despite the air warming trend is not the largest in JJA.

Done. We agree with you. Some positive feedbacks, such as the sea ice-albedo feedback, are the most sensitive in JJA to any small change in the atmosphere above as sea ice reaches to the minimum state in this season. Thus, these feedbacks can be easily triggered and further allows more open ocean to uptake solar radiation. We added more detailed explanation on this in the revision in lines 91-94 as below:

“This downward heat transfer suggests that recent fall upper ocean warming (Fig. 1a) originates from more absorption of heat at the surface in summer (i.e, JJA) because sea ice-albedo feedback which is more efficient in summer allows stronger oceanic uptake of solar radiation during ice melting seasons.”

3. Page 5, is it proper to put Laptev Sea in PPSS instead of APSS?

We agree with you. We put the Laptev Sea in the APSS in the revision.

4. Page 6, between atmospheric warming and ocean, change to “atmospheric and ocean ...”

Corrected.

5. Page 7, “but this is a secondary effect from the reduction of multiple reflection resulting from a shrinkage of sea ice coverage.” Although references are provided, it is still better to provide a few sentences explaining it directly inside this paper, as it is important to know the reason that you neglect this relatively large budget term.

The decreasing of downwelling shortwave radiation is mainly due to the reduction of multiple reflection between sea ice and clouds as sea ice is melted. We rewrote this statement and made this clearer in the revision in lines 162-165 as below:

“Downwelling shortwave radiation (DSR) decreases substantially over the 40 years, but this is a secondary effect resulting from the reduction of multiple reflection between a shrinkage of sea ice coverage and clouds⁴²⁻⁴⁴.”

6. Page 8, “the warming in the nudging simulations does not advance as far northward as in the observations.” How about the recent 2 decades? See the major comments.

Please see the response above.

Page 9, “...are from a decades-long spin up which generates warmer states than the real condition in 1979. This means that our estimate of the role of internal variability could in fact be underestimated by a warm bias in the models’ initial states...” How big is the artificial trend (numerical drift in temperature)? If you repeat the runs for 40 years, using exactly the same setup as the wind nudging runs, just without applying wind nudging, do you get warming or cooling trend in the upper Arctic Ocean? Without answering this question, the original sentence in the paper does not make much sense.

Before we ran our nudging simulations, we had a 150 years spin-up perpetual run (continuously forced by reanalysis winds of year 1979 in the Arctic) to let the model have a sufficient long time to adjust to reanalysis winds in the Arctic that may be very different from the model's own wind fields in the Arctic. The simulated upper ocean temperature in this spin-up appears to be stabilized after almost 100 years (Fig. R4) and then levels off in the last 50 years. In this way, we think we have minimized any possible numerical drift that is caused by adding winds in the model. Then, the last 5 years (Jan/1) are used as initial states of our nudging runs, in which reanalysis winds are allowed to vary from 1979 to 2018. These initial states are warmer than the real conditions in 1979. However, we agree with you that it remains unclear how this initial warming bias impacts our estimation of the following temperature trends driven by imposed varying winds from 1979 to 2018. So we remove this sentence in the revision.

Response Fig. 4. The Arctic Ocean domain-average upper (0 – 50 m average) ocean temperature (°C) in SON during 150-yr spin-up perpetual run. The domain used for the calculation here is the area enclosed by the solid black contour in **Fig. 1b**. (Supplementary Fig. 11 in the manuscript)

Page 10, capture over – capturing over

Corrected.

Page 10, in turn reduces – in turn increases?

Corrected. “reduces” should be “increases”.

Page 10, last paragraph, Figure 4 – Figure 3

Corrected.

Page 11, “This suggests that the wind-driven atmospheric circulation change plays an important role in driving POHT through the Bering Strait” This statement can be well supported by citing literature (Danielson2014, Zhang2020):

Danielson, S. L., Weingartner, T. J., Hedstrom, K. S., Aagaard, K., Woodgate, R., Curchitser, E., & Stabeno, P. J. (2014). Coupled wind forced controls of the Bering-Chukchi shelf circulation and the Bering Strait throughflow: Ekman transport, continental shelf waves, and variations of the Pacific-Arctic sea surface height gradient. *Progress in Oceanography*, 125, 40-61.

Zhang, W., Wang, Q., Wang, X., & Danilov, S. (2020). Mechanisms driving the interannual variability of the Bering Strait throughflow. *Journal of Geophysical Research: Oceans*, 125, e2019JC015308.

Done. These two relevant references have been cited in the revision.

Page 11, “but that POHT changes from the Atlantic Ocean are likely determined by other factors, such as the initial ocean condition, deeper layer oceanic variability, internal oceanic thermohaline variability in the Arctic and heat transport from the lower latitudes where observed winds are not specified.” The complication about explaining the variability of Atlantic water inflow can be well supported by citing related literature, for both Fram Strait and Barents Sea branches (e.g., Muilwijk2019, Wang2019, Wang2020):

Muilwijk, Morven, Ilicak, Mehmet, Cornish, Sam B., Danilov, Sergey, Gelderloos, Renske, Gerdes, Rüdiger, Haid, Verena, Haine, Thomas W. N., Johnson, Helen L., Kostov, Yavor, Kovács, Tamás, Lique, Camille, Marson, Juliana M., Myers, Paul G., Scott, Jeffery, Smedsrud, Lars H., Talandier, Claude and Wang, Qiang (2019) Arctic Ocean response to Greenland Sea wind anomalies in a suite of model simulations, *J. Geophys. Res.-Oceans*, 124, 6286-6322.

Wang, Q., Wang, X., Wekerle, C., Danilov, S., Jung, T., Koldunov, N., Lind, S., Sein, D., Shu, Q., Sidorenko, D. (2019). Ocean heat transport into the Barents Sea: Distinct controls on the upward trend and interannual variability, *Geophysical Research Letters*, 46, 13180-13190.

Wang, Q., Wekerle, C., Wang, X., Danilov, S., Koldunov, N., Sein, D., Sidorenko, D., von Appen, W. and Jung, T. (2020) Intensification of the Atlantic Water supply to the Arctic Ocean through Fram Strait induced by Arctic sea ice decline, *Geophysical Research Letters*, 47, e2019GL086682.

Done. The references have been updated with these new and important publications.

Page 2 where Arctic Ocean warming is mentioned, or Page 12-13, where Atlantification is mentioned, a recent very relevant paper is not cited:

Qi Shu, Qiang Wang, Zhenya Song, Fangli Qiao (2021). The poleward enhanced Arctic Ocean cooling machine in a warming climate. *Nature Communications*, 12, 2966.

Thanks. The reference has been added.

Methods

Poleward ocean heat transport. Page 16, the transect of Atlantic Water inflow seems to be not a closed transect.

The transect of Atlantic Water inflow is shown in blue contours in the stereographic projection in Fig. 4a. To avoid misleading, we adjusted the Atlantic Gate to stay along 76°N from 15°W to 60°E. In our study, the gateway is consistent with the region we focused on, that circled by climatological JJA sea ice area. The gateway for the Atlantic one is a closed transect, and its two ends are landlocked by Greenland and Novaya Zemlya. We also modified the descriptions for two gates in the revision in lines 458-461 as below:

“Cross sections are indicated by blue and greens contours in the stereographic projection in Fig. 4a for the Atlantic Ocean gateway (15°W - 60°E longitude at 76°N latitude) and the Pacific Ocean gateway (170°E - 160°W longitude at 70°N latitude).”

Figure 4 caption, “ten wind nudging experiments”, is it ten?

Corrected. “ten” should be “five”.

All figures: figure quality needs to be better in the final version

Done. All original plots in the eps format will be uploaded if needed.

Supplementary figure 5, the color range for temperature is different for the first and third row. Any reason?

The first row exhibits the temperature in February, but the third row is that in August. We used different color ranges to better visualize different magnitude of mean states of temperature in the two months. But we use the same color scale for ORAS5 and the nudging runs in each month.

The paper is largely well written, but proofreading is still needed.

A thorough proofreading is done.

Reference:

Zhang, J., Rothrock, D. A. & Steele, M. Warming of the Arctic Ocean by a strengthened Atlantic Inflow: Model results. *Geophysical Research Letters* **25**, 1745–1748 (1998).

Reviewers' Comments:

Reviewer #1:

Remarks to the Author:

The revised manuscript is well presented. I commend the authors for their work which largely addressed my main concerns. I only have two major and a few minor comments. I recommend publication subject to minor revisions.

Main comments:

1) L34 and L91: Sea ice-albedo feedback. On my last reading I was wondering about the exact role of sea ice loss in the proposed causal chain. The authors state that the upper ocean warming is driven by the "sea ice-albedo feedback induced by atmospheric dynamics". To my understanding, the sea ice-albedo feedback is the specific process whereby the retreat of sea ice leads to increased heat absorption by the ocean surface which in turn **drives increased sea ice melt**. As far as I understand the story presented here (particularly my reading of L91 and following), the reduction of sea ice is driven by atmospheric processes, and the reduction of sea ice leads to increased upper ocean warming. I am not clearly seeing the "feedback" part here, whereby the increasingly warm upper ocean drives further sea ice melt. I would rather call this an albedo "effect" of replacing a lighter surface with a darker one.

2) The revised explanation of Z300 on L107 is somewhat helpful, although it is not clear to me from the current version how this is linked to changes in anticyclonic circulation. In their response file, the authors write that "In our case, we suggest that Z300 is changing towards a stronger anticyclonic circulation over Greenland and the Arctic Ocean (increases in Z300) over the past four decades." However, in the manuscript the term "anticyclonic" is only featured once (L 303), where it is stated that the circulation trend is toward stronger anticyclonic conditions, but no explanation of its relation to Z300 is given. In my opinion this warrants some further explanation.

Minor points:

L 37 : I suggest removing "accelerated", it's more confusing than insightful

L 48 : I'd remove "energy"

L 61 : I find the second part of this sentence "particularly during the years 2007-2012" somewhat confusing - the process either explains 40% of the trend or it doesn't, no?

L 93 : "...because THE sea ice-albedo ..."

L 150 : remove "field"

L 248 : What exactly is the difference between the shortwave components of Qnet (DLR and USR) and Qshort_bl ? I was under the assumption that DLR and USR are computed at the ocean surface? And I wonder how this is different to Qshort_bl, which is the shortwave radiation heat flux into the ocean boundary layer - i.e., how is the flux into the ocean surface different than into the ocean boundary layer?

Reviewer #2:

Remarks to the Author:

The authors have done a good job at thoroughly addressing the points raised in the previous revision round. I only have a few final, and very minor, comments:

1) Because your heat transports for the Pacific and the Atlantic gateway now are calculated relative to different reference temperatures (-1.9°C and 0°C, respectively), the time series in Figure 4 are no longer directly comparable. The information about the different reference temperatures used should therefore be stated in the Figure 4 caption in addition to the methods section.

2) In the author response you state that you are not quite sure why there is a mismatch in the magnitude of the Atlantic and Pacific gateway heat transports in ORAS5 versus the nudged run. I think checking the time series for 0-50m temperature and volume transport in ORAS5, CESM, and the nudged run will give you at least some idea of the source, and it should be easy to do. E.g., is the inflow temperature higher or the inflow volume transport stronger in ORAS5 compared to in CESM? Is there a known cold bias in the CESM model system? The magnitude difference is quite glaring, and I think it deserves a mention in the text.

3) L.199-201: I think 'reproduces' is a little strong here seeing that the magnitude of the spatial pattern is quite different. The difference in itself is not a problem, as you're not trying to explain 100% of the trend in ORAS5.

4) L.283-284: It's the variability that is captured, not the magnitude - so be clear on this.

5) L.350-552: When describing an ocean reanalysis or state estimate it is common to state which observational products are used in the assimilation + the atmospheric forcing used. From what I can tell: ERA-Interim atmospheric forcing, EN4 for subsurface temperature and salinity. Is HadISST used for assimilating SST and SIC? I think these details should be clear from reading your 'Reanalysis and observational data' section.

6) Has ORAS5 been evaluated for the Arctic before or is this the first time it's been used this far north? If there are known biases in ORAS5 in the Arctic, this should be stated. I'm guessing not many TS profiles goes into the EN4 product in the central Arctic (and thus as constraints in ORAS5). This should be mentioned as a clear caveat of your analysis.

Reviewer #3:

Remarks to the Author:

The revised version addressed most of my concerns.

I have only one major comment on this version. Lines 246-259 were added during the revision in response to the comment of another reviewer. I find this paragraph (and the associated new plot) does not add any new content to the paper. Instead, it causes confusion. The question was about the physical processes responsible for the time lag at depth. The added text does not answer the question. The quantity Q_{short_bl} is also not clearly defined. Furthermore, the location intended to explain the processes is also not logical. It should be placed where Figure 1e-h is described.

Actually, the explanation is simple, so no new analysis is really needed. Reviewer2 gave some hints. Increased mixing can simply explain the deepening of the warming signals, while increasing sea ice coverage in fall brings surface temperature back to freezing temperature, which explains losing correlation at the surface.

In addition, in Figure 3h, the deepening of the high correlation signal is too much compared to ORAS5 in Figure 1. This can be also linked to vertical mixing --- in the numerical model, vertical mixing is overestimated (either parameterized mixing, or the numerical mixing), which is a common issue to nearly all models in the community (e.g., Ilicak2016). The too deep and thick Atlantic Water layer in the simulation shown in Supplementary Fig5 similarly indicates too much vertical mixing in the model. I would suggest to remove the added text and plot, and just add short explanations at proper places.

A few minor comments

164 between a shrinkage of sea ice coverage – between a shrinking sea ice coverage

173 the second “CESM1” – which

251 exists -- exist

265 to drive – in driving

265 Which regions do you refer to with “there”?

L274-275 In Figure S8b, the ORAS5 curve does not show a trend. Is the plot correct? The nudging runs have a trend in the current plot.

L279: a high correlation alone does not prove a strong influence for the central Arctic Ocean. In the revised version, I still did not see convincing mechanism linking Atlantic heat with central Arctic upper ocean temperature without time lag. The mechanisms should be mentioned explicitly in the paper, if you have convincing ones.

L285 wind-driven atmospheric circulation change plays – winds play

At the end of Section “Poleward ocean heat transport contribution”, a summary sentence about Atlantic POHT is missing. Just an example:

“Importantly, the fact that our nudging simulations did not capture an upward trend in POHT through the Atlantic Gate further supports our main finding based on the nudging simulations, that is, the role of summertime ‘atmospheric’ processes in expediting Arctic Ocean warming.”

That is, the fact that the nudging simulations did not produce POHT trend makes your interpretation easier and is supportive.

L326 write differently for “..., and ..., and”

Some of the citations in the text are not updated after the reference list was extended. Please check. One example is in line 333.

We really appreciate that the all three reviewers gave our manuscript a very careful and thorough read and review. We also would like to thank the reviewers for the time they have invested in reviewing the first and second versions of the paper and for their new constructive comments. These comments and suggestions continuously helped improve the presentation and clarify our main points. We have incorporated the reviewer's comments and suggestions into the revised manuscript and extend our appreciation to the three reviewers for providing us many insightful comments and suggestions in the acknowledgment. The main changes we incorporated in this revision include more discussion about how Q_{short_bl} is simulated in the model and how the mixed layer depth changes in ORAS5 and our nudging experiments over the period, more details of original data sources assimilated into ORAS5 and potential limitations of these data sources. We also clean up all minor problems pointed out by the reviewers. Below are our detailed replies to each of his/her comments. We hope that we have sufficiently addressed all comments in the revised manuscript. In the responses, words in black are the original reviews, followed by our replies in blue.

Reviewer #1 (Remarks to the Author):

The revised manuscript is well presented. I commend the authors for their work which largely addressed my main concerns. I only have two major and a few minor comments. I recommend publication subject to minor revisions.

Main comments:

1) L34 and L91: Sea ice-albedo feedback. On my last reading I was wondering about the exact role of sea ice loss in the proposed causal chain. The authors state that the upper ocean warming is driven by the “sea ice-albedo feedback induced by atmospheric dynamics”. To my understanding, the sea ice-albedo feedback is the specific process whereby the retreat of sea ice leads to increased heat absorption by the ocean surface which in turn **drives increased sea ice melt**. As far as I understand the story presented here (particularly my reading of L91 and following), the reduction of sea ice is driven by atmospheric processes, and the reduction of sea ice leads to increased upper ocean warming. I am not clearly seeing the “feedback” part here, whereby the increasingly warm upper ocean drives further sea ice melt. I would rather call this an albedo “effect” of replacing a lighter surface with a darker one.

Actually, the interpretation of the term “feedback” really depends on the time scale we focus on. Within a summer, a strong sea ice melting over an area in a short period (say, a week) will increase the SW absorption of the upper ocean there, which in turn will favor more sea ice melting in the following weeks in and around the same area. The reason to use “feedback” in our original version is to imply this type of relative fast process as this term is commonly used by many literatures. But I think these two terms are interchangeable here and both well reflect the key concept we want to express. So, we agree to use “effect” in the revision.

2) The revised explanation of Z300 on L107 is somewhat helpful, although it is not clear to me from the current version how this is linked to changes in anticyclonic circulation. In their response file, the authors write that “In our case, we suggest that Z300 is changing towards a stronger anticyclonic circulation over Greenland and the Arctic Ocean (increases in Z300) over the past four decades.” However, in the manuscript the term “anticyclonic” is only featured once (L 303), where it is stated that the circulation trend is toward stronger

anticyclonic conditions, but no explanation of its relation to Z300 is given. In my opinion this warrants some further explanation.

We agree. Since most readers of this article will probably be more familiar with oceanographic phenomena, we should give a little more background on what Z300 and anticyclonic circulation refer to respectively. In the North Hemisphere, a high-pressure anomaly (with higher Z300 or warmer air from the surface to the pressure level at 300 hPa) can be also called as an anticyclone circulation that comes with clockwise winds blowing around its center. Therefore, in our paper, we sometimes describe this process as a circulation change toward a higher Z300 or a stronger anticyclonic condition. We have added one sentence to explain the relationship between Z300 and anticyclonic circulation in the revision in lines 108-110 as below:

“a higher Z300 also means that the circulation changes toward a pattern with stronger anticyclonic movement in the Northern Hemisphere”

Minor points:

L 37 : I suggest removing “accelerated”, it’s more confusing than insightful

Removed as suggested.

L 48 : I’d remove “energy”

Removed as suggested.

L 61 : I find the second part of this sentence “particularly during the years 2007-2012” somewhat confusing - the process either explains 40% of the trend or it doesn’t, no?

Ding et al. (2019) did not calculate the contribution of internal variability during those six years specifically (2007 – 2012), so we cannot give an accurate number for that period. We have removed the second part of that sentence in the revision to emphasize what has been really done in the cited study.

L 93 : “...because THE sea ice-albedo ...”

Added as suggested.

L 150 : remove “field”

Removed as suggested.

L 248 : What exactly is the difference between the shortwave components of Q_{net} (DLR and USR) and Q_{short_bl} ? I was under the assumption that DLR and USR are computed at the ocean surface? And I wonder how this is different to Q_{short_bl} , which is the shortwave radiation heat flux into the ocean boundary layer - i.e., how is the flux into the ocean surface different than into the ocean boundary layer?

Yes, the shortwave components of Q_{net} (DSR and -USR) are radiative fluxes computed at the surface. The variable Q_{short_bl} refers to the solar short-wave heat flux absorbed by the

boundary layer, which is calculated by the KPP vertical mixing scheme in CESM1's POP2 component (Roekel et al., 2018). In the scheme, Q_{short_bl} is jointly determined by the boundary layer depth, intensity of surface net shortwave flux, and a vertical shortwave absorption function (Roekel et al., 2018). The first step in the KPP scheme is to calculate the boundary layer depth (about 30 m in the Arctic Ocean in our model) by searching for the shallowest depth where the bulk Richardson number is equal to a critical value. Once this depth is determined, fraction of solar short-wave flux penetrating to the bottom of this layer can be calculated on the basis of exponential decay in Jerlov water type or another empirical transmission function considering chlorophyll amount. This fraction is about ~ 6% at the depth of 30 m. Then Q_{short_bl} , representing how much of incoming shortwave flux at the surface is to be absorbed by the whole depth of the boundary layer, can be calculated by a simple formula: $(1-6\%) \times$ net incoming SW at the surface. This variable is useful for us to quantify how much SW can penetrate into the entire boundary layer due to the surface albedo effect in this study. To make it clearer, we have added some explanations about Q_{short_bl} in the revision in lines 448-453 as below:

“In addition, one advantage of this set of simulations is that the POP2 of the CESM1 can simulate Q_{short_bl} , which is calculated by the KPP vertical mixing scheme and not available in ORAS5. This variable will tell us how much of incoming shortwave flux at the surface can be absorbed by the whole depth of the ocean surface boundary layer in each oceanic grid based on the fraction of solar short-wave flux penetrating to the bottom of this layer⁷⁷.”

Reference:

Ding, Q. et al. Fingerprints of internal drivers of Arctic sea ice loss in observations and model simulations. *Nature Geoscience* **12**, 28–33 (2019).

Van Roekel, L. et al. The KPP Boundary Layer Scheme for the Ocean: Revisiting Its Formulation and Benchmarking One-Dimensional Simulations Relative to LES. *Journal of Advances in Modeling Earth Systems* **10**, 2647–2685 (2018).

Reviewer #2 (Remarks to the Author):

The authors have done a good job at thoroughly addressing the points raised in the previous revision round. I only have a few final, and very minor, comments:

1) Because your heat transports for the Pacific and the Atlantic gateway now are calculated relative to different reference temperatures (-1.9°C and 0°C , respectively), the time series in Figure 4 are no longer directly comparable. The information about the different reference temperatures used should therefore be stated in the Figure 4 caption in addition to the methods section.

Yes. We have added the statement about the different reference temperatures used for the ocean heat transports through the Pacific Ocean and Atlantic Ocean gateways in Figure 4's caption.

2) In the author response you state that you are not quite sure why there is a mismatch in the magnitude of the Atlantic and Pacific gateway heat transports in ORAS5 versus the nudged run. I think checking the time series for 0-50m temperature and volume transport in ORAS5, CESM, and the nudged run will give you at least some idea of the source, and it should be easy to do. E.g., is the inflow temperature higher or the inflow volume transport stronger in ORAS5 compared to in CESM? Is there a known cold bias in the CESM model system? The magnitude difference is quite glaring, and I think it deserves a mention in the text.

Thanks for pointing this out. We have added the Fig. R1 as Fig. S14 in the revision to explain the mismatch in the magnitude of POHT through the Atlantic and Pacific Gates in ORAS5 versus the nudging runs and CESM-LEN. As shown in Fig. R1c&d, the poleward volume transports through the Atlantic and Pacific Gates are stronger in ORAS5 compared to those in CESM-LEN and the nudging runs. This appears to be the main cause of the mismatch, and this bias could be due to the fact that the CESM has a difficulty in simulating the mean state over the region. Another way to clearly see the effect of volume transport in causing the mismatch is to substitute temperatures or velocities from the nudging runs to ORAS5, as shown in Fig. R2. We found that POHT calculated in this hybrid way using ORAS5 velocities and simulated temperatures (red line) looks more closer to POHT in ORAS5 (blue line). This further supports that underestimated poleward volume transport in the model should be the culprit of the mismatch. We have modified the discussion of the underlying cause of mismatch and moved it to the Method part in the revision in lines 496-499 as below:

“We also note that there is a mismatch between the magnitudes of simulated POHT and that in ORAS5 through the two gateways (Fig. 4d&e). The main source of this mismatch appears to be the underestimated intensity of poleward volume transport in the model (Supplementary Fig. 14d&e).”

Response Fig. 1. a&b. SON cross section-average upper (0 – 50 m average) ocean temperature (°C) along the Atlantic Ocean gateway in **a**, and along the Pacific Ocean gateway in **b** from the ensemble average of the five wind nudging experiments (red line), the CESM-LEN 40 members average (green line) and the ORAS5 reanalysis (blue line) for the period 1979 – 2018. **c&d.** The same as **a&b** but for the poleward volume transport (Sv) in the upper 50 m in SON. (Supplementary Fig. 14 in the manuscript)

Response Fig. 2. SON upper 50 m POHT (TW) from 1979 to 2018 through the Atlantic Ocean gateway in **a**, and through the Pacific Ocean gateway in **b** using different combinations of temperatures and velocities from ORAS5 and the nudging runs.

3) L.199-201: I think ‘reproduces’ is a little strong here seeing that the magnitude of the spatial pattern is quite different. The difference in itself is not a problem, as you’re not trying to explain 100% of the trend in ORAS5.

We agree. We have reworded that sentence in the revision to better describe the results by saying that their patterns are similar rather than the magnitude.

4) L.283-284: It’s the variability that is captured, not the magnitude – so be clear on this.

Done. We have modified that sentence in the revision by emphasizing that the model can capture the variability instead of the magnitude.

5) L.350-552: When describing an ocean reanalysis or state estimate it is common to state which observational products are used in the assimilation + the atmospheric forcing used. From what I can tell: ERA-Interim atmospheric forcing, EN4 for subsurface temperature and salinity. Is HadISST used for assimilating SST and SIC? I think these details should be clear from reading your 'Reanalysis and observational data' section.

ORAS5 uses reprocessed SST from HadISST2 and OSTIA operational, reprocessed sea-ice concentration from OSTIA and OSTIA operation, reprocessed in-situ profiles from EN4 with XBT/MBT correction and GTS operational in the assimilation. Global mean sea-level changes in ORAS5 are constrained using AVISO DT2014 L4 MSLA + NRT. In addition, ORAS5 is produced with the atmospheric forcing using ERA-40 (before 1979), ERA-Interim (1979 - 2015), and ECMWF NWP (2015 - present). We have modified the 'Reanalysis and observational data' section to include more details about which observational products and the atmospheric forcing are used in creating ORAS5 in the revision.

6) Has ORAS5 been evaluated for the Arctic before or is this the first time it's been used this far north? If there are known biases in ORAS5 in the Arctic, this should be stated. I'm guessing not many TS profiles goes into the EN4 product in the central Arctic (and thus as constraints in ORAS5). This should be mentioned as a clear caveat of your analysis.

We agree that reanalysis data have their limitations, especially around the Arctic due to short duration and poor coverage of field and satellite observations. However, ORAS5 has been used successfully in several studies of the Arctic Ocean climate. Carton et al. (2019) evaluated different reanalyses including ORAS5 with observations by examining two types of Arctic climate variability: the changes in Atlantic Water properties in the Nordic seas on the Atlantic side and the changes in Beaufort Gyre freshwater storage on the Pacific side, and they found that ORAS5 shows reassuringly similar Atlantic Water variability in the Greenland, Iceland, and Norwegian Seas, including the transition from cool and fresh conditions in the mid-1990s to warmer and saltier conditions after 2000. Recently published paper by Shu et al. (2021) also evaluated the ORAS5 by comparing its ocean temperature, salinity, and upper 100 m ocean heat content with long-term observations in the Barents Sea, and they concluded that ORAS5 can be used for investigating climate change in the Barents Sea region. But we should be also aware that there is a limited amount of TS profiles used in the EN4 product in the central Arctic Ocean because that the Argo buoys cannot operate under perennial sea ice. We have added one sentence to point out this limitation of ORAS5 in reflecting real observations in the central Arctic Ocean in the revision in lines 374-378 as below:

“Although various in-situ observations are assimilated in ORAS5, there is only a limited amount of temperature/salinity profiles in the EN4 product in the central Arctic Ocean as the buoys cannot operate under perennial sea ice, which may potentially degrade the accuracy of ORAS5 in reflecting real observations over the region.”

Reference:

Carton, J., Penny, S. & Kalnay, E. Temperature and salinity variability in soda3, ECCO4r3, and ORAS5 ocean reanalyses, 1993-2015. *Journal of Climate* **32**, (2019).

Shu, Q., Wang, Q., Song, Z. & Qiao, F. The poleward enhanced Arctic Ocean cooling machine in a warming climate. *Nat Commun* **12**, 2966 (2021).

Reviewer #3 (Remarks to the Author):

The revised version addressed most of my concerns.

I have only one major comment on this version. Lines 246-259 were added during the revision in response to the comment of another reviewer. I find this paragraph (and the associated new plot) does not add any new content to the paper. Instead, it causes confusion. The question was about the physical processes responsible for the time lag at depth. The added text does not answer the question. The quantity Q_{short_bl} is also not clearly defined. Furthermore, the location intended to explain the processes is also not logical. It should be placed where Figure 1e-h is described.

The main point of the paragraph about Q_{short_bl} is to illustrate a JJA atmospheric-driven mechanism to explain upper ocean warming in the following SON. Please also see our new response to Reviewer #1 (the last question, page #2-3) regarding the detailed procedure of CESM1 in calculating this variable (Q_{short_bl}). In short, Q_{short_bl} represents how much of incoming shortwave flux at the surface is to be absorbed by the whole depth of the boundary layer, which is about 30m in and around the central Arctic Ocean. This variable is useful to quantify how much SW can penetrate into the entire boundary layer due to the surface albedo effect. In the revision, we have added more explanations about the physical meaning of this variable (Q_{short_bl}). Since it can be only simulated by CESM1 but not the ORAS5 reanalysis. It is more appropriate to discuss this process when we start to talk about the nudging experiment.

Actually, the explanation is simple, so no new analysis is really needed. Reviewer2 gave some hints. Increased mixing can simply explain the deepening of the warming signals, while increasing sea ice coverage in fall brings surface temperature back to freezing temperature, which explains losing correlation at the surface.

In addition, in Figure 3h, the deepening of the high correlation signal is too much compared to ORAS5 in Figure 1. This can be also linked to vertical mixing --- in the numerical model, vertical mixing is overestimated (either parameterized mixing, or the numerical mixing), which is a common issue to nearly all models in the community (e.g., Ilicak2016). The too deep and thick Atlantic Water layer in the simulation shown in Supplementary Fig5 similarly indicates too much vertical mixing in the model.

I would suggest to remove the added text and plot, and just add short explanations at proper places.

We agree with the reviewer on this point and have added some relevant discussions in the revision in lines 247-254 about the importance of increasing vertical mixing and variations of the mixed layer depth (MLD) in contributing to the deepening of upper ocean warming in SON. We also examined the model's skill in simulating the observed MLD by calculating the climatology and trend of MLD in SON from ORAS5 and the wind nudging experiments (Fig. R3). It seems that the wind nudging experiments capture the long time increasing trend of MLD in SON as observed in ORAS5, especially that during the period 2000 to 2018, although its magnitude is slight smaller than ORAS5. The reviewer is correct that the vertical mixing in the numerical model is overestimated to some degree, and this can be clearly seen in Fig. R3a&d, showing that the simulated mean MLD in the Arctic Ocean is slightly deeper than that from ORAS5. We also mentioned this point in the revision. However, we decide to keep the discussions about Q_{short_bl} since this discussion is useful to explain not only the lead lag relationship between atmospheric circulation and ocean warming, but also the

physical linkage between surface Q_{net} and temperature warming in the upper ocean. A physical understanding of this linkage is expected by Reviewer #2 in the first round of review.

Response Fig. 3. **a.** Climatology (m) of the mixed layer depth in SON from the ORAS5 reanalysis for the period 1979 – 2018. **b-c.** Linear trend (m per decade) of the mixed layer depth in SON for the period 1979 - 2018 in **b**, and for the period 2000 - 2018 in **c** from the ORAS5 reanalysis. **d-f.** The same as **a-c**, but from the ensemble average of the five wind nudging experiments. Black stippling in all plots indicates statistically significant trends at the 95% confidence level. (Supplementary Fig. 7 in the manuscript)

A few minor comments

164 between a shrinkage of sea ice coverage – between a shrinking sea ice coverage

Corrected as suggested.

173 the second “CESM1” – which

Corrected as suggested.

251 exists – exist

Corrected as suggested.

265 to drive – in driving

Corrected as suggested.

265 Which regions do you refer to with “there”?

“there” means “in the Arctic, especially over the APSS”. We have made this point clearer in the revision

L274-275 In Figure S8b, the ORAS5 curve does not show a trend. Is the plot correct? The nudging runs have a trend in the current plot.

Thanks for catching this error. The colors and labels of lines were inadvertently swapped in the old Fig. S8, and we have replotted this figure with correct labels and colors.

L279: a high correlation alone does not prove a strong influence for the central Arctic Ocean. In the revised version, I still did not see convincing mechanism linking Atlantic heat with central Arctic upper ocean temperature without time lag. The mechanisms should be mentioned explicitly in the paper, if you have convincing ones.

Yes, we agree with the reviewer that it is difficult to understand a strong influence of ocean heat transport via the Atlantic Gate in causing simultaneous warming over the central Arctic Ocean just based on the high correlation alone in Fig. 4b. As we stated in the revision in lines 289-292, there could be an air-sea heat flux exchange that quickly takes the warm signal from the Barents Sea via the atmosphere and communications it to the central Arctic Ocean (Zhang et al., 1998). In addition, an oceanic pathway may be also responsible for the establishment of this simultaneous connection in SON. To test this possibility, we calculated the correlations of MAM POHT with upper ocean temperature in JJA and SON and correlations of POHT in JJA and SON with SON upper ocean temperature (Fig. R4) and found that high simultaneous correlations in SON in the central Arctic Ocean may also reflect some influences originating from the Barents Sea in preceding JJA or MAM. Due the long persistent nature of oceanic processes, the SON connection presented in Fig. 4b could be due to a third factor that occurs earlier and can influence both SON POHT and upper ocean temperature in the central Arctic Ocean. We also briefly talk about this speculation in the revision in lines 292-294. However, we still leave it as an open question and believe further detailed analysis with some modeling work is needed to comprehensively understand the mechanisms linking SON Atlantic ocean heat transport with central Arctic upper ocean temperature in SON in the future.

Response Fig. 4. a&b. Correlations of MAM (Mar-Apr-May) upper 50 m POHT through the Atlantic Ocean gateway with upper ocean temperature field in SON in **a**, and in JJA in **b**. **c&d.** Correlations of JJA (in **c**) and SON (in **d**) upper 50 m POHT through the Atlantic Ocean gateway with upper ocean temperature field in SON.

L285 wind-driven atmospheric circulation change plays – winds play

Changed as suggested.

At the end of Section “Poleward ocean heat transport contribution”, a summary sentence about Atlantic POHT is missing. Just an example:

“Importantly, the fact that our nudging simulations did not capture an upward trend in POHT through the Atlantic Gate further supports our main finding based on the nudging simulations, that is, the role of summertime ‘atmospheric’ processes in expediting Arctic Ocean warming.” That is, the fact that the nudging simulations did not produce POHT trend makes your interpretation easier and is supportive.

Yes, thanks for your suggestion. We agree with you and have added this very well-written summary sentence to end that section in the revision in lines 312-315 as suggested.

L326 write differently for “..., and ..., and”

Corrected.

Some of the citations in the text are not updated after the reference list was extended. Please check. One example is in line 333.

Thanks for giving our manuscript a very careful read. Yes, the citations in that line was not updated synchronously. We have updated all the citations and carefully checked them to make sure all citations are correct.

Reference:

Zhang, J., Rothrock, D. A. & Steele, M. Warming of the Arctic Ocean by a strengthened Atlantic Inflow: Model results. *Geophysical Research Letters* 25, 1745–1748 (1998).

Reviewers' Comments:

Reviewer #1:

Remarks to the Author:

I commend the authors on a further improved version of this manuscript and believe that my main comments have been largely addressed.

I still had several more minor queries and a substantial number of copy-editing comments which I hope will be helpful (see the annotated pdf attached). I also would urge the authors to go over the text once more carefully and tighten up and clarify where possible as I still struggled at points throughout.

Once this has been done I will be happy to recommend this work for publication.

Reviewer #3:

Remarks to the Author:

I am happy to see that all my comments have been properly addressed. I would suggest to accept the paper for publication.

Reviewer #1 (Remarks to the Author):

I commend the authors on a further improved version of this manuscript and believe that my main comments have been largely addressed.

I still had several more minor queries and a substantial number of copy-editing comments which I hope will be helpful (see the annotated pdf attached). I also would urge the authors to go over the text once more carefully and tighten up and clarify where possible as I still struggled at points throughout.

Once this has been done I will be happy to recommend this work for publication.

We really appreciate that the reviewer gave our manuscript a very careful and thorough read and review again. We have accepted all your editorial suggestions. Please find our point-to-point responses to your main concerns below. Line numbers provided below are according to that in the Main text_revision_clean_version file.

L72: what does its refer to here? atmospheric internal variability? maybe rephrase?

‘its’ refers to the interaction of atmospheric internal variability and upper ocean temperature mentioned in the previous sentence. We have rephrased this sentence in the revision in lines 71-73 as below:

“We seek a physical understanding of the underlying mechanism of this atmosphere-ocean interaction and a quantification of its contribution to the recent warming compared with that due to anthropogenic forcing.”

L97-98: lead-lag between what entities? this is a little unclear

The lead-lag relationship is examined between air temperature and ocean temperature. We have rewritten this sentence in the revision in lines 95-98 as below:

“This connection operates well on interannual time scales, with the causal direction examined by a lead-lag relationship between JJA Pan-Arctic tropospheric (surface to 300 hPa) average air temperature and ocean temperature in each month and depth (Fig. 1g&h).”

L164: it's confusing that USR is discussed in terms of its negative value. I would just have it positive and then just have the trend in fig 2e negative etc

We agree. We removed all the negative signs of USR and ULR and mentioned that all radiative flux variables are positive downward in the Methods part and figure captions in the revision.

L254-257: both reanalyses and GCMs are notoriously bad at faithfully capturing the MLD and its variability in the Arctic. Maybe worth commenting on this?

We agree. We have added one sentence in lines 250-253 in the revision to point out this uncertainty associated with the MLD and its variability in reanalysis and simulations.

“Although the vertical mixing is overestimated in the simulations compared with that in ORAS5 (Supplementary Fig. 7a&d; Fig. 3h), it remains unclear which one is closer to observations since most climate models and reanalyses contain large uncertainties to capture the MLD and its variability in the Arctic ⁴⁹.”

L331-333: This appears to be in contrast with Fig 3c&e, no?

The aim of this sentence was to mention that the simulated upper ocean warming for the period 1979 to 2018 is mostly confined to the Chukchi, East Siberian, and Laptev Seas. To avoid a possible confusion, we have rephrased this sentence to make it clearer in the revision in lines 322-326 as below:

“Our nudging experiments confirm that adiabatic dynamical forcing associated with winds in the Arctic is able to explain up to 24% of SON upper ocean warming from 1979 to 2018, which is mostly confined to the Chukchi, East Siberian, and Laptev Seas, and up to ~ 60% of the Pan-Arctic upper ocean warming for the period 2000 to 2018.”

L336: use of long hyphens (--) and short hyphens (-), as well as spaces before and after is inconsistent

Thanks for pointing this out. We have carefully checked all the hyphens in the revision to make sure that we correctly use them throughout the paper.

L363: why italicized (also later on you hyphenate it)

We have removed the italicized style and hyphens for “in situ” in the revision.

L271: what resolution?

The horizontal resolution of ORAS5 is approximately $0.25^\circ \times 0.25^\circ$, and we have added it in the revision.

L419: why are these different latitudes?

The aim of using the latitude 60-90°N is that large scale atmospheric forcing reflected by Z300 over 60-70°N are also important for upper ocean warming in the Arctic considering its impacts in driving atmospheric and ocean heat transport from the lower latitudes toward the Arctic. Actually, the MCA using Z300 within 70-90°N also shows a very similar pattern with what we present now.

L427: why not use the very commonly used W/m^2

Done. We have changed all the units of radiation fluxes to W/m^2 in the revision.

L431-433: <https://eos.org/opinions/parentheses-are-not-for-references-and-clarification-saving-space>

We have reorganized this sentence and removed all the parentheses to make it clearer in the revision in lines 422-425 as below:

“where DLR and ULR are downwelling and upwelling longwave radiations, respectively; DSR and USR are downwelling and upwelling shortwave radiations, respectively; SHF is sensible heat flux; LHF is latent heat flux. All radiative flux variables are positive downward and represent heat transferred from the atmosphere to the surface.”

Figure 3: Why are the ORAS5 and CESM ocean temps (y-axes ranges) so different? It would be helpful to comment on this.

The bias of ocean temperature magnitudes between ORAS5 and CESM could be due to that fact that the CESM has a difficulty in simulating the mean state over the region. In the section “The mean sea ice and oceanic states simulated by wind-nudging experiments” in the Methods, we have compared the long-term mean ocean temperature in ORAS5 and the wind nudging experiments over a vertical-cross section, and the results show that the model well captures the main patterns of the Arctic Ocean. However, it is true that ORAS5 has a warmer mean state of upper ocean than that in the simulations. We have emphasized this point in the Methods part of the revision in lines 470-473 as below:

“However, deficiencies still remain: One is the absence of a warm summer Pacific Water layer in the Canada Basin in the model and the other is the cooler mean state of upper ocean temperature in the model. These are common issues even in some regional models⁶².”

Reviewer #3 (Remarks to the Author):

I am happy to see that all my comments have been properly addressed. I would suggest to accept the paper for publication.

Thank you for agreeing to accept our paper for publication.